

# Investigating the yield of $H_2O$ and $H_2$ from methane oxidation in the stratosphere

Franziska Frank[1], Patrick Jöckel[1], Sergey Gromov[2,3], and Martin Dameris[1]

[1]Deutsches Zentrum für Luft- und Raumfahrt (DLR), Institut für Physik der Atmosphäre, Oberpfaffenhofen, Germany
[2]Max-Planck-Institute for Chemistry, Air Chemistry Departement, Mainz, Germany
[3]Institute of Global Climate and Ecology Roshydromet & RAS (IGCE), Moscow, Russia

*Correspondence to:* Franziska Frank (franziska.frank@dlr.de)

**Abstract.** An important driver of climate change is stratospheric water vapour (SWV), which in turn is influenced by the oxidation of atmospheric methane ($CH_4$). In order to parameterize the production of water vapour ($H_2O$) from $CH_4$ oxidation, it is often assumed that the oxidation of one $CH_4$ molecule yields exactly two molecules of $H_2O$. However, this assumption is based on an early study, which also gives evidence, that this is not true at all altitudes.

In the current study we re-evaluate this assumption with a comprehensive systematic analysis using a state-of-the art Chemistry-Climate model (CCM), namely the ECHAM/MESSy Atmospheric Chemistry (EMAC) model, and present three approaches to investigate the yield of $H_2O$ and hydrogen gas ($H_2$) from $CH_4$ oxidation. We thereby make use of Module Efficiently Calculating the Chemistry of the Atmosphere (MECCA) in a box model and global model configuration. Furthermore, we use the kinetic chemistry tagging technique (MECCA-TAG) to investigate the chemical pathways between $CH_4$, $H_2O$ and $H_2$, by
being able to distinguish hydrogen atoms stemming from $CH_4$ and other sources.

We apply three approaches, which all agree that assuming a yield of 2 overestimates the production of $H_2O$ in the lower stratosphere (calculated as 1.5–1.7). Additionally, transport and subsequent photochemical processing of longer-lived intermediates raise the local yield values in the upper stratosphere and lower mesosphere above 2 (maximum > 2.2). In the middle and upper mesosphere, the influence of loss and recycling of $H_2O$ increases, making it a crucial factor in the parameterization of the yield
of $H_2O$ from $CH_4$ oxidation. An additional sensitivity study with the Chemistry As A Boxmodel Application (CAABA) shows a dependence of the yield on the hydroxyl radical (OH) abundance. No significant temperature dependence is found. We focus representatively on the tropical zone between 23° S-23° N, where seasonal variations are negligible. It is found in the global approach that presented results are mostly valid for mid latitudes as well.

Our conclusions question the use of a constant yield of $H_2O$ from $CH_4$ oxidation in climate modeling and encourage to apply
comprehensive parameterizations that follow the vertical profiles of the $H_2O$ yield derived here and take the chemical $H_2O$ loss into account.

## 1   Introduction

It is beyond question that water vapour ($H_2O$) is an important greenhouse gas (GHG). The current study focuses on stratospheric water vapour (SWV), which is by itself an influential driver of climate change. SWV, for example, induces a reduction of





stratospheric ozone concentration (Stenke and Grewe, 2005; Revell et al., 2016), cools the stratosphere (Revell et al., 2012; Forster and Shine, 1999; Maycock et al., 2014) and produces a positive radiative forcing (Solomon et al., 2010). Changes in SWV are mainly driven by troposphere-stratosphere exchange (e. g. through deep convection in the tropics (Fueglistaler and Haynes, 2005)). However, there is also a chemical contribution to SWV, mostly by oxidation of methane ($CH_4$) and

hydrogen gas ($H_2$). These gases are still abundant above the tropopause to act as significant in-situ photochemical sources of $H_2O$. Besides $H_2O$, $CH_4$ is a powerful GHG as well, with a 34 times higher climate effect than an equivalent amount of carbon dioxide ($CO_2$) on a time horizon of 100 years (IPCC, 2013). It also introduces secondary climate effects through the additional SWV. The strong linkage of $CH_4$ and SWV represents a decisive factor of the net climate effect of $CH_4$. Enhanced $CH_4$ concentrations are likely expected in the future Earth's atmosphere and can impact the otherwise rather dry stratosphere

substantially (Rohs et al., 2006).

Nevertheless, to account for the contribution of $CH_4$ to SWV, in current climate modeling it is common either to use a Chemistry-Climate model (CCM) with a complex chemistry set up, which puts high demands on computational resources, or a General Circulation model (GCM) or Chemical Transport model (CTM) with – if at all – a parameterization of the chemical sources of SWV. A parameterization of the chemical feedback onto SWV requires to estimate the yield of $H_2O$ from $CH_4$

oxidation, which is defined as the production of $H_2O$ per oxidized $CH_4$ molecule. A common simple assumption of the yield of $H_2O$ from $CH_4$ oxidation is that one oxidized $CH_4$ molecule produces two $H_2O$ molecules in the stratosphere. This simple parameterization is based on a first estimation of the $H_2O$ yield from $CH_4$ oxidation, using a simplified methane chemistry without chlorine in a two dimensional photochemistry model (le Texier et al., 1988).

This is a widely accepted approximation (Myhre et al., 2007; Stowasser et al., 1999) and is also affirmed by aircraft ob-

servations, which state that $2 \cdot [CH_4] + [H_2O]$ (also named as the total stratospheric hydrogen budget) is fairly constant in the stratosphere being 6.8-7.6 ppmv (Hurst et al., 1999; Rahn et al., 2003; Dessler et al., 1994; Stowasser et al., 1999). Although this suggests that all atomic hydrogen (H) from $CH_4$ oxidation reaches $H_2O$, it must be noted that the referenced observation studies do not distinguish, whether the H in $H_2O$ comes from $CH_4$ or from $H_2$, which also originates from the troposphere. Thus, calculations based on observed mixing ratios show a net production of $H_2O$ only, but not the yield of $H_2O$ specifically

from $CH_4$ oxidation (Hurst et al., 1999). Furthermore, $H_2$ mixing ratios, when measured as well, show an almost absent vertical gradient, which can be explained by the supposition that the $H_2$ sink is in photochemical equilibrium with its production from $CH_4$ oxidation. Hence, all additional $H_2$ by $CH_4$ is leveled by the oxidation of $H_2$ and balances the $2 \cdot [CH_4] + [H_2O]$ and $H_2$ content in the stratosphere (Rahn et al., 2003). Nevertheless, Hurst et al. (1999) took the weak anti-correlation of $H_2$ and $CH_4$ into account and calculated a net production of $H_2O$ over loss of $CH_4$ of 1.973 ±0.003, differing from the assumed value of 2,

which would be the case if all H goes into $H_2O$.

Still, for reasons of simplification, several GCMs use the approximation that the yield of $H_2O$ from $CH_4$ oxidation is exactly two (Monge-Sanz et al., 2013; ECMWF, 2007; Austin et al., 2007; Boville et al., 2001; Mote, 1995; Eichinger et al., 2015). In the ECHAM/MESSy Atmospheric Chemistry (EMAC) model (Jöckel et al., 2010), for example, explicitly configured in a CTM-like set-up without interactive chemistry, the production of SWV from $CH_4$ oxidation is calculated in a simplified way





using a specifically introduced CH$_4$ tracer (by applying the CH4 submodel) according to:

$$\frac{d}{dt}[H_2O] = -\gamma_{H_2O} \cdot \frac{d}{dt}[CH_4] \tag{1}$$

with $\gamma_{H_2O} = 2$ as the yield of H$_2$O.

However, this approximation first and foremost neglects the chemical loss of H$_2$O (mostly by reaction with excited oxygen (O($^1$D)) and by photolysis). Using this parameterization, SWV is solely added and not removed by chemistry. Moreover, the results of le Texier et al. (1988) also suggest that the yield of H$_2$O from CH$_4$ oxidation is not exactly two, accounting for the part of H diverted into H$_2$ production and that the share of H$_2$ increases at higher altitudes. Therefore, following the results of le Texier et al. (1988) precisely, we would generate a certain bias by using a yield of 2 in Eq. (1), especially at higher altitudes, where 2·[CH$_4$]+[H$_2$O] approx. const. does not hold anymore. In the mesosphere, for example, the loss of H$_2$O becomes increasingly relevant, shifting the balance between H$_2$O and H$_2$ towards the latter. Furthermore, the net production calculated by Hurst et al. (1999) and the yield of le Texier et al. (1988) also do not agree well in the lower stratosphere, which can indeed be explained by the indistinguishable inputs from H$_2$ and CH$_4$ oxidation in observations as stated before. Yet, this does also indicate that the yield from CH$_4$ oxidation itself must be even lower than suggested by the net production, which is calculated based on observations. It is, therefore, questionable, if the assumption of $\gamma_{H_2O} = 2$ for the CH$_4$ oxidation is indeed applicable.

In this study we re-evaluate the findings of le Texier et al. (1988) with multiple approaches using a modern CCM with a complex state-of-the-art chemistry mechanism. Our goal is to assess the currently used assumption of the constant yield as in Eq. 1 with $\gamma_{H_2O} = 2$ and investigate, if a parameterization solely based on CH$_4$ is sufficient to reproduce the chemical yield of H$_2$O from CH$_4$ oxidation. As an additional remark, it should be noted that difficulties with yield estimates can be expected especially in the stratosphere, as it is not as well mixed as the turbulent troposphere.

We show three approaches to determine the yield of H$_2$O from CH$_4$ oxidation. The first two approaches use the kinetic chemistry tagging technique (MECCA-TAG, Gromov et al. (2010)), either (1) in a box model set-up with the Chemistry As A Boxmodel Application (CAABA, Sander et al. (2011a)) and (2) in a global simulation, with the EMAC (Jöckel et al., 2010) model. For the third approach (3), we rely on the assumption that the hydrogen budget in the stratosphere is conserved, mostly consisting of fractions of H, H$_2$, H$_2$O and CH$_4$.

We apply MECCA-TAG (Gromov et al., 2010) in all approaches to run a comprehensive chemistry setup, while being able to track the production of H$_2$O originating explicitly from CH$_4$ oxidation. A conceptionally different approach would be the extended Crutzen's sequential method used by Johnston and Kinnison (1998) to estimate the gross ozone loss by CH$_4$. Despite that this study focuses on the tropospheric and lower stratospheric ozone (O$_3$), it is a practical example on the derivation of atmospheric trace gas yields. By applying MECCA-TAG, however, it is not necessary to explicitly write down the chemical net reactions as this is done in the extended Crutzen's sequential method.

The paper is structured as follows: In section 2 we present the methods and theoretical background of our studies, followed by the results in section 3. Section 4 comprises a detailed discussion and section 5 summarizes the findings and gives an outlook for further studies.



## 2  Methods

### 2.1  The model set-up

#### 2.1.1  EMAC

The applied global chemistry climate model is EMAC, which is a state-of-the art numerical chemistry and climate simulation
system that includes sub-models describing tropospheric and middle atmosphere processes and their interaction with oceans,
land and human influences (Jöckel et al., 2010). It uses the second version of the Modular Earth Submodel System (MESSy)
to link multi-institutional computer codes. The core atmospheric model is the 5th generation European Centre Hamburg general circulation model (ECHAM5) (Roeckner et al., 2006). For the global simulations in the present study we applied EMAC
(ECHAM5 version 5.3.02, MESSy version 2.53.0) in the T42L90MA-resolution, i.e. with a spherical truncation of T42 (corresponding to a quadratic Gaussian grid of approx. 2.8 by 2.8 degrees in latitude and longitude) with 90 vertical hybrid pressure
levels up to 0.01 hPa. The applied model setup comprises particularly the submodels MECCA (Module Efficiently Calculating the Chemistry of the Atmosphere) (Sander et al., 2005) and MECCA-TAG (kinetic chemistry tagging technique) (Gromov
et al., 2010).

The  MECCA represents the chemical core of EMAC. The applied chemistry is based on a chemical mechanism, which,
for example, was already used for the base simulations in the Earth System Chemistry integrated Modelling (ESCiMo) project
(Jöckel et al., 2016). The mechanism is extended to resolve specific intermediates in the $CH_4 \rightarrow H_2O$ reaction chain, resulting
in slightly more comprehensive chemical kinetics. The full chemical mechanism is part of the supplement.

#### 2.1.2  The kinetic tagging technique MECCA-TAG

MECCA-TAG (Gromov et al., 2010) enables the user to tag certain elements, without modifying the underlying standard chemical mechanism. It can either be applied for simulating isotopologues of selected trace gases or used to investigate elemental
exchange between the species of interest. For example, a model study was carried out with focus on the carbon and oxygen
isotope composition of carbon monoxide (CO) (Gromov et al., 2010).

In the current study we use the tagging technique (in the so called fractional mode) to investigate the pathways of H atom
transfer from the source $CH_4$ to $H_2O$ via all simulated intermediates. In order to do so, we create counterparts of the species
of interest (e.g., those containing H) in an isolated doubled set of studied reactions (e.g., $CH_4$ oxidation chemistry) in the
same chemical mechanism simulated. By doing so, we are able to quantify the fraction of molecules (hence their H content)
stemming from $CH_4$ oxidation only, as well as their production and loss rates, which are used for the yield calculations.
Furthermore, we improve the latter by quantifying the H, which is recycled in the given reactions.

In this particular case, we count the $H_2O$ molecules created from $CH_4$ oxidation pathways and are able to distinguish the H
from $CH_4$ from the H of other sources ($H_2$, NMHCs, HCFCs etc.). However, those that further break down to other HOx compounds (and subsequently produce $H_2O$ again) are counted separately. Overall, such an approach is the "online" approximation
of the technique used by Lehmann (2004) and helps to avoid double-counting issues in yield derivation. Ultimately, we are





able to quantify the fraction of H atoms populating the species of the complete ($CH_4 \rightarrow H_2O/H_2 \leftrightarrow HO_x$)-cycle, including their fractions recycled via $H_2O$.

### 2.1.3 CAABA

For the photochemical box model studies we use the Chemistry As A Boxmodel Application (CAABA) in model version 3.0
(Sander et al., 2011a). CAABA equipped with MECCA (CAABA/MECCA) provides an atmospheric chemistry box model, simulating single air parcels with the chemical mechanism identical to that used in EMAC. CAABA/MECCA is, moreover, using the MESSy interface to attach certain submodels to the box model system. The used submodels in the current study, in addition to MECCA, are SEMIDEP (applies deposition fluxes) and JVAL (calculates photolysis rates) (Sander et al., 2014).

CAABA simulates one box at one pressure and temperature specific for a given latitude and altitude in the atmosphere. To derive a pseudo vertical profile of the yield, 35 independent boxes superimposed upon each other at the equator are simulated with prescribed conditions following a standard atmosphere profile ((NOAA/NASA, 1976) accessed via https://www.digitaldutch.com/atmoscalc (digital dutch, 1999)). The equator is chosen for its negligible seasonal cycle. Since the boxes represent different temperature and pressure levels and therefore distinct chemical regimes throughout the middle atmosphere, it is possible to illustrate the vertical dependence of the yield.

Note that the purpose of the box model simulation is to demonstrate the steady state conditions expected at different altitudes. In order to do so, we mimic the effect of vertical transport between the boxes by prescribing the vertical distribution of the relevant species concentrations for:

1. $CH_4$ and all species acting as in-situ sources of H (primarily NMHCs and HCFCs), which are not produced in the chemical mechanism,

2. long-lived substances, such as $NH_3$ and $N_2O$,

3. $N_2$ and $O_2$, whose mixing ratios are virtually constant throughout the considered altitude range,

4. NO and $O(^1D)$, to constrain the HOx-NOx-cycle to the given initial state

5. $SO_2$, Cl and Br, for the same reason as in 4. with respect to ClOx, BrOx and sulfate compounds,

6. $H_2O$ and $H_2$ mixing ratios and therefore serving as a H sink for the limitless influx of H via the fixed source species (indicated in 1.).

Other species, particularly the hydroxyl radical (OH) and hydroperoxyl ($HO_2$), are unconstrained in the simulations unless otherwise noted. All initial mixing ratios of the chemical species are taken from a climatology over the years 2000–2010 of the RC1SD-base-10 EMAC simulation of the ESCiMo project (Jöckel et al., 2016).

Because a priori fractions of H from $CH_4$ (or tagged H) in the species of the chemical mechanism is not known, all tagged species are initialized with zero. The simulation of every box is run for 200 years to make sure that all tagged species have filled up to a steady state.



## 2.2 Calculation of the chemical $H_2O$ yield from $CH_4$ oxidation

A straight forward definition of the direct yield is the ratio of the production of $H_2O$ molecules by the loss of $CH_4$, as depicted in Eq. (2).

$$\gamma_{H_2O}^{direct}(CH_4) = \frac{\mathbf{P}_{H_2O}^{I}}{\mathbf{L}_{CH_4}} \tag{2}$$

5      with variables listed in Table 1. The yield of $H_2O$ from the oxidation of $CH_4$ ($\gamma_{H_2O}$) represents the units of molecule $H_2O$ per molecule $CH_4$ (i.e. [molecule/molecule]) and is displayed dimensionless throughout this work.

The loss of $CH_4$ ($\mathbf{L}_{CH_4}$) in MECCA includes the reactions with OH, $O(^1D)$ and Cl, as well as photolysis (see Reactions (R1) - (R6)).

$$CH_4 + O(^1D) \quad \rightarrow \quad CH_3 + OH \tag{R1}$$

$$\rightarrow \quad CH_3O + H \tag{R2}$$

$$\rightarrow \quad CH_2O + H_2 \quad ^{a,} \tag{R3}$$

$$CH_4 + OH \quad \rightarrow \quad CH_3 + H_2O \quad ^{a,} \tag{R4}$$

$$CH_4 + Cl \quad \rightarrow \quad HCl + CH_3 \quad ^{a,} \tag{R5}$$

$$CH_4 + h\nu \quad \rightarrow \quad products \quad ^{b,} \tag{R6}$$

15      with reaction rates of a, from Sander et al. (2011b) and photolysis rate of b, calculated by JVAL (Sander et al., 2014).

Following these reactions, H atoms from $CH_4$ are distributed among intermediates (not shown) and eventually reach $H_2O$. Produced $H_2O$ reacts further and gets removed, by reactions (R7) and (R9).

$$H_2O + O(^1D) \quad \rightarrow \quad 2OH \quad ^{a,} \tag{R7}$$

$$SO_2 + OH + O_2 + H_2O \quad \rightarrow \quad H_2SO_4 + HO_2 \quad ^{a,} \tag{R8}$$

$$H_2O + h\nu \quad \rightarrow \quad H + OH \quad ^{b,} \tag{R9}$$

with reaction rates of a, from Sander et al. (2011b) and photolysis rate of b, calculated by JVAL (Sander et al., 2014).

In consecutive reactions H is again recycled into $H_2O$. The direct yield calculated by Eq. (2) represents the $H_2O$, which is produced in the chemical mechanism and directly emerges from $CH_4$ oxidation. However, this is not the additional $H_2O$ of the whole chemical process. It also cannot be used in a simplified set-up for the methane chemistry and the production

25      of SWV parameterized as by Eq. (1), because no chemical depletion of water is considered. Hence, we suggest to define the effective yield of $H_2O$, which takes into account that water is recycled in consecutive reactions and that recycled water is again destroyed. The process is sketched in Fig. 1. During this recycling process, some H is converted to species other than $H_2O$, filling up to a steady state or leaving the HOx-cycle once and for all. The effective yield is therefore always equal to or smaller than the direct yield in a closed system.



**Table 1.** Variable names as used in Equations 2, 3 and 4.

| name | description |
| --- | --- |
| $\mathbf{L}_{CH_4}$ | loss of $CH_4$ molecules |
| $\mathbf{P}^I_{H_2O/H_2}$ | direct production of $H_2O/H_2$ by H from $CH_4$ |
| $\mathbf{L}^I_{H_2O/H_2}$ | loss of directly produced $H_2O/H_2$ |
| $\mathbf{P}^{II}_{H_2O/H_2}$ | production of recycled $H_2O/H_2$, hence the H already has been part of a $H_2O/H_2$ produced by $CH_4$ |
| $\mathbf{L}^{II}_{H_2O/H_2}$ | loss of recycled $H_2O/H_2$ |
| $\mu_{H_2O/H_2}$ | lost $H_2O/H_2$ during the recycling |

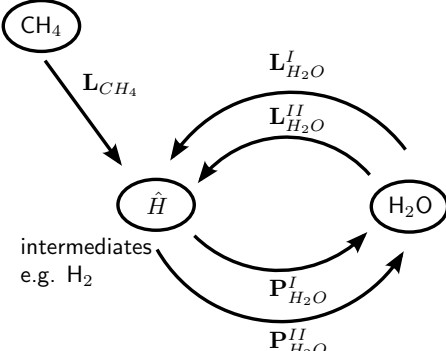

**Figure 1.** Sketch on the production and recycling of $H_2O$.

We define the effective yield of $H_2O$ in this study as in Eq. (3), with $\mu$ accounting for the lost $H_2O$, due to subsequent loss and recycling of $H_2O$ molecules:

$$\gamma^{eff}_{H_2O}(CH_4) = \frac{\mathbf{P}^I_{H_2O} - \mu_{H_2O}}{\mathbf{L}_{CH_4}} \quad \text{with} \quad \mu_{H_2O} = \mathbf{L}^I_{H_2O} + \mathbf{L}^{II}_{H_2O} - \mathbf{P}^{II}_{H_2O} \tag{3}$$

Variables are listed in Table 1.

5   Due to the implementation of the tagging technique, counting of recycled H (as described in section 2.1.2) can only be applied with respect to one species at once. Hence, the effective yield can only be calculated either for $H_2O$ or $H_2$ in the same simulation. Similar to that for $H_2O$, recycling of $H_2$ is calculated in the chemical mechanism, that is, the recycled H is counted as soon as it is leaving $H_2$. The corresponding formula for $H_2$ is derived similarly to Eq. (3) and reads as follows:

$$\gamma^{eff}_{H_2}(CH_4) = \frac{\mathbf{P}^I_{H_2} - \mu_{H_2}}{\mathbf{L}_{CH_4}} \quad \text{with} \quad \mu_{H_2} = \mathbf{L}^I_{H_2} + \mathbf{L}^{II}_{H_2} - \mathbf{P}^{II}_{H_2} \tag{4}$$





The chemical conversion from $CH_4$ to $H_2O$ follows some intermediate reactions. Hence, the loss of $CH_4$ and the eventual production of $H_2O$ do not occur simultaneously.

Furthermore, in reality, chemistry undergoes diurnal variations. The major changes occur during daylight. At night, virtually no photo-sensitive chemistry takes place, which results in very low OH concentrations. This reduces $CH_4$ loss and $H_2O$ production to a nighttime-low. A diurnal average smoothes the difference between day and night to a representative value. This is based on the assumption that the system is in a quasi-steady-state. A quasi-steady-state implies that equal integral production and loss are simulated throughout a given time interval, e.g. a day, a month or a year. Monthly $\gamma_{H_2O}$ averages, as presented in this study, which average over the simulated diurnal cycle, are sufficient for the application of a simplified $CH_4$ loss/$H_2O$ production rates calculation with prescribed monthly varying OH distributions.

For these reasons, we apply in our analysis Eq. (3) to annual averages of the production and sink terms simulated in the boxes representing conditions typical for the tropics, where in addition seasonal variations are negligible. In the global simulations with EMAC we calculate an average over zonally averaged tropical bands.

In the following we compare the direct and effective yields of $H_2O$ and $H_2$ from $CH_4$ oxidation obtained in simulations with the box model and EMAC.

# 3 Results

## 3.1 Box model approach

### 3.1.1 Simulation with unrestrained oxidation capacity

The direct and the effective yield of $H_2O$ from $CH_4$ oxidation of the box model approach (i.e. simulation Exp1), calculated as indicated in Eq. (2) and Eq. (3) respectively, are shown as a pseudo vertical profile in Fig. 2 by 35 vertically stacked boxes following the standard atmosphere at the equator. The shown results comprise also boxes on tropospheric levels. However, since the physical water cycle (e.g. evaporation, clouds) exceeds the influence of the $CH_4$ oxidation onto $H_2O$, the kinetic production of $H_2O$ is irrelevant in the troposphere. All values below the tropopause level (approximately 100 hPa in the tropics) are therefore not part of the analysis presented in this work.

The direct yield in Fig. 2 (left) is 1.7 around the tropopause and increases monotonically up to 2 at 4 hPa. It remains constant until 0.2 hPa, where it starts to decrease monotonically down to about 0.65 at the uppermost layer.

The direct and the effective yields do not differ significantly for water vapor throughout the stratosphere and most of the mesosphere. This suggests, that the $H_2O$ recycling at these pressure levels and chemical regimes is predominant and all broken down water is regenerated. Nevertheless, in the mesosphere at approx. 0.1 hPa, the effective yield decreases more strongly than the direct yield, reaching the minimum of 0.17 at 0.02 hPa, with a slight increase to 0.39 at the topmost layer at 0.01 hPa.

The value of 2 between 4 and 0.2 hPa reflects that all H from $CH_4$ reaches $H_2O$ eventually at these altitudes, supporting the assumption as accepted in the literature. In the lower stratosphere and upper mesosphere, however, the box model results show that assuming a yield of 2 will lead to an overestimated $H_2O$ production.





The yield of $H_2$ (see Fig. 2 (right)) shows a mostly anti-correlated behavior with respect to the yield of $H_2O$. Throughout most of the stratosphere the effective and direct yields of $H_2$ differ by about 0.2, while the effective yield drops down to 0 between 4 and 0.2 hPa, i.e. exactly in the region where the yield of $H_2O$ attains its maximum. In accordance with the decreasing yield of $H_2O$, the direct and effective yields of $H_2$ increase substantially at higher altitudes, giving evidence that more and more H becomes diverted to and stays in $H_2$ instead of continuing towards $H_2O$.

A good indicator for the rate of general chemical reactivity in the atmosphere is the $CH_4$ lifetime, which is mostly influenced by both, temperature, and the concentration of the reaction partners. The lifetime of $CH_4$ ($\tau_{CH_4}$) with respect to its sinks OH, chlorine (Cl), O($^1$D) and photolysis is defined as:

$$\tau_{CH_4} = \frac{1}{(k_{OH}*[OH] + k_{Cl}*[Cl] + k_{O1D}*[O1D])*c_{air} + j_{CH_4}} \tag{5}$$

with $k_X$ being the reaction rate coefficients of $CH_4$+X in [cm3 s$^{-1}$], [X] being the mixing ratio of species X, $c_{air}$ the concentration of dry air in [molecules cm$^{-3}$] and $j_{CH_4}$ the photolysis rate of $CH_4$ in [molecules s$^{-1}$].

The area, where the $H_2O$ yield attains its maximum, i.e. where it is 2, corresponds to the area, where the lifetime of $CH_4$ attains its stratospheric minimum (see Fig. 3). However, the $CH_4$ lifetime does not fully explain the behavior of the chemical yield, since in the upper mesosphere both, yield and lifetime, drop to a minimum, which can be explained by the emerging role of photolysis in this area. This further suggests that OH is an important factor in the $H_2O$ yield in the stratosphere, but does not influence it alone. It becomes replaced by photolysis in the mesosphere, which influences the $CH_4$ lifetime and, more importantly, destroys $H_2O$ and initiates its recycling.

A sensitivity study concerning the impact of OH onto $\gamma_{H_2O}$ is presented in the next section.

### 3.1.2 Sensitivity with respect to OH

The results of the previous section revealed that the effective yield of water vapor from $CH_4$ oxidation depends on the box location, hence the chemical regime at a certain pressure level. Particularly, OH is one of the major oxidants that largely controls the conversion of $CH_4$ to $H_2$ and $H_2O$ respectively.

In the simulations shown above (Exp1) the OH is unconstrained, however, its final (equilibrated) OH concentration does not deviate much from the initial values (see Fig. 4).

In further sensitivity simulations with CAABA, OH is initialized with the reference from EMAC multiplied with constants and kept constant throughout the simulation. This introduces an additional prescribed hydrogen carrying species, which introduces or withdraws hydrogen to or from the system. However, contribution of OH to the total H abundance in the system was found negligible. The first four simulations reduce the OH concentration by the factors of 0.5 (SS1), 0.1 (SS2), 0.05 (SS3) and 0.01 (SS4) respectively, while the fifth one is performed with a doubled OH concentration (SS5). One additional simulation represents the reference simulation (Ref), which started with an OH concentration identical to the analysis above, except that OH is kept constant. The simulations are listed in Table 2. The simulation set-up uses extreme perturbations of the OH concentration to provide a qualitative estimate of the impact of OH onto the $H_2O$ yield from $CH_4$ oxidation.




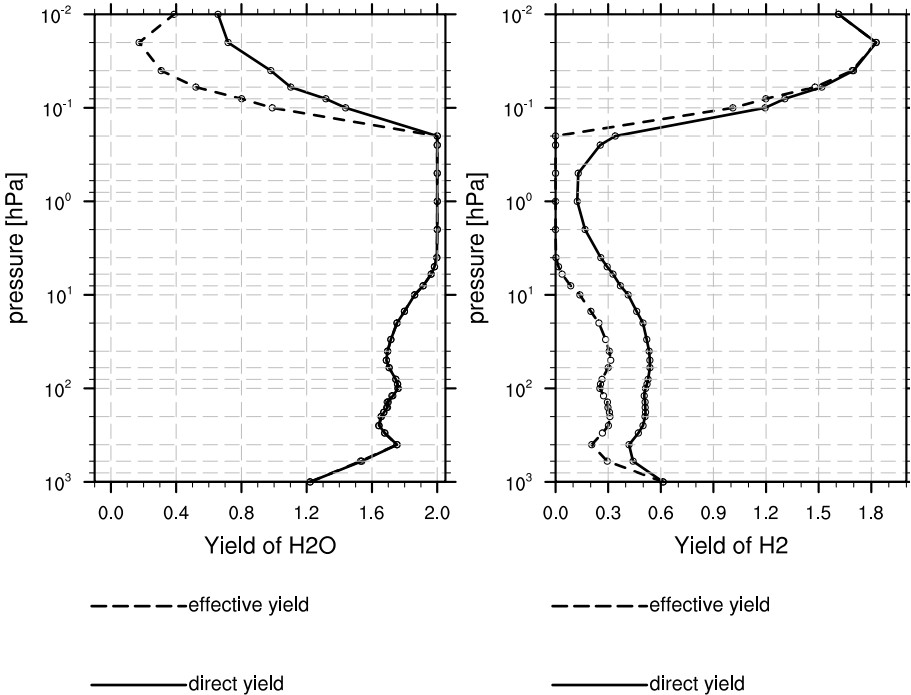

**Figure 2.** The pseudo vertical profile shows the $H_2O$ yield (left) and $H_2$ yield (right), calculated by the box model approach. The solid line represents the direct yield, the dashed line represents the effective yield and circles indicate the pressure levels of the model boxes.

**Table 2.** Overview of simulations carried out in this study, including box model simulations and the sensitivity study concerning $H_2O$ yield dependence on OH as well as the global simulations with EMAC.

| Name | description |
| --- | --- |
| Exp1 | Experiment with unconstrained OH |
| Ref | Reference with standard fixed OH concentration from yearly climatology of RC1SD-base-10 |
| SS1 | Sensitivity simulation with 0.5×OH from Ref |
| SS2 | Sensitivity simulation with 0.1×OH from Ref |
| SS3 | Sensitivity simulation with 0.05×OH from Ref |
| SS4 | Sensitivity simulation with 0.01×OH from Ref |
| SS5 | Sensitivity simulation with 2.0×OH from Ref |
| Exp2 | Global simulation with EMAC, MECCA and MECCA-TAG |





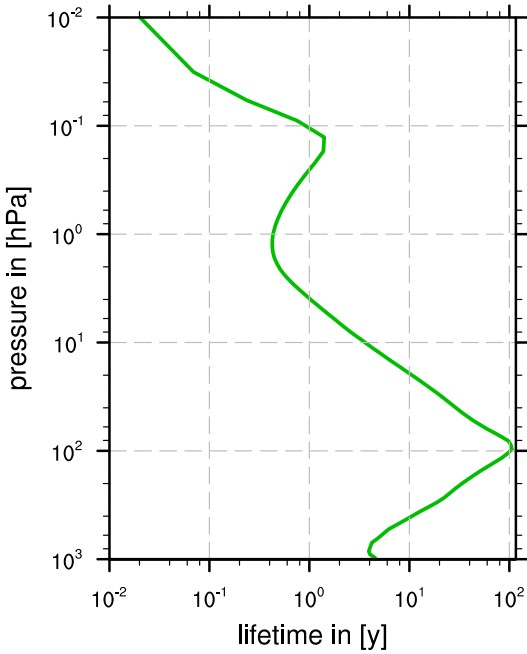

**Figure 3.** Vertical profile of $CH_4$ lifetime in the tropics with respect to removal by OH in years.

The results of the sensitivity simulations are shown in Fig. 5. First of all, the initial experiment Exp1 (see Fig. 2) and the reference experiment of the sensitivity study Ref (see Fig. 5 red line), show mostly consistent results compared to each other concerning the effective and direct yield, which confirms that prescribing OH is adequate. However, in the upper mesosphere, where the OH concentration has the largest difference (cf. Fig. 4), the effective yield in the experiment Ref drops already at 1

5  hPa significantly. Additionally, the effective yield in the experiment Ref reaches a value lower than the effective yield in the experiment Exp1 in this area. Nevertheless, the direct yield is not considerably different between these two experiments. This once more supports the assumption of a strong OH dependence of the $\gamma_{H_2O}$.

Comparing experiment Ref with SS1 shows that reducing the OH concentrations by half reduces the direct and effective yields by about 0.05 in the lower stratosphere. Altogether, the direct yield profiles are rather similar in experiment Ref and

10  SS1, with an exception of lower values in SS1 within the 10–1 hPa range and above 0.2 hPa. Prominent, however, is the difference in the effective yield. In the experiment SS1 the effective yield drops to zero already at 0.04 hPa and does not have the local enhancement seen in experiment Ref around 0.2–0.02 hPa.

Considering the sensitivity simulations SS2–SS4, the effect of OH reduction on $\gamma_{H_2O}$ becomes more apparent. The effective yield drops to zero already above 60 hPa. The direct yield shows strongly reduced values in the stratosphere, with a local

15  minimum at 20 hPa for SS2 and SS3 and a bit above for SS4, being 1.08, 0.92 and 0.78 respectively. Above 20 hPa the direct yield increases towards a local maximum at 2 hPa, following the profile of the $CH_4$ lifetime. Above 2 hPa the direct yield decreases nearly monotonically.







**Figure 4.** Reference OH concentration in the tropics from ESCiMo experiment RC1SD-base-10a (purple) and OH concentration as simulated in respective boxes (red).

In the experiment SS5, with doubled OH, $\gamma_{H_2O}$ is about 0.07 higher compared to experiment Ref and nearly replicates the results of experiment Exp1 in the mesosphere, where the OH equilibrated at a value of about twice that of the reference OH concentration from EMAC.

Compared to the yields of $H_2O$, the effective and direct yields of $H_2$ show moderate dependence on OH concentration. The yield of $H_2$ is rather constant at lower levels, reaches its minimum around the stratopause and increases again above that to its maximum. Around the stratopause and in the lower mesosphere all experiments show similar results. In lower boxes the simulations with lower OH show higher yields and vice versa. In contrast to this, the boxes in the middle mesosphere and above show an inverted behavior. Except, however, for experiment SS5, which results in a lower yield than in the reference simulation.

Moreover, profiles of yield of $H_2$ from the oxidation of $CH_4$ ($\gamma_{H_2}$) of experiments SS2, SS3 and SS4 overall do not vary much compared to each other.

To investigate the vertical profile of the effective yield of $H_2O$ in more detail, all terms of Eq. (3) are plotted separately in Fig. 6 for the experiment Ref. In line with the $CH_4$ lifetime, we see that loss of $CH_4$ and production of $H_2O$ minimize around the tropopause and maximize close to the stratopause. The maximum of the primary loss of $H_2O$ in the stratosphere is



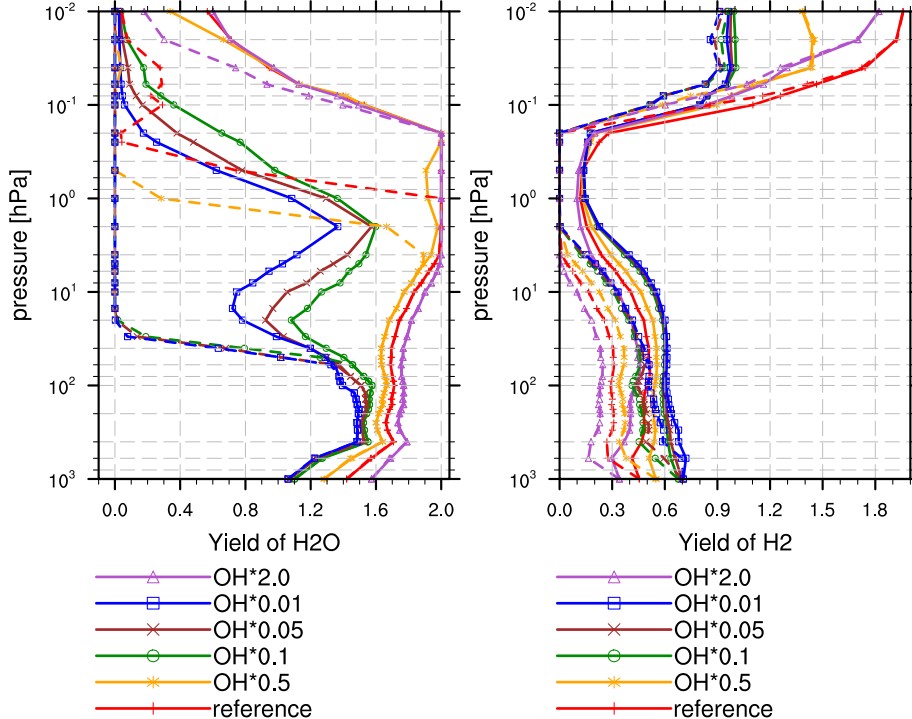

**Figure 5.** Pseudo vertical profiles of the $H_2O$ yield (left) and $H_2$ yield (right), calculated by the box model approach. Solid lines represent the direct yield, the dashed lines the effective yield and circles indicate the pressure levels of the model boxes. OH concentrations are prescribed in all simulation to the initial values of the respective vertical box. The plot shows simulations with the reference OH concentration (Ref, red, plus signs) as well as the OH concentration times 2 (SS5, purple, triangles), times 0.5 (SS1, orange, asterisks), times 0.1 (SS2, green, circles), times 0.05 (SS3, brown, crosses) and times 0.01 (SS4, blue, squares).

slightly shifted vertically. Above the stratopause, the recycling of $H_2O$ becomes more important. This is indicated by increased secondary loss and production of $H_2O$ and is further reflected by the reduced effective yield in the mesosphere.

Summarizing, reduction of the OH concentrations leads to a proportionally larger decrease in the $H_2O$ yield at higher altitudes owing to the differences in the chemical regimes. On the other hand, increasing the OH concentration also increases

5    the direct and foremost the effective yield of $H_2O$.

The results of the sensitivity study suggest that the effective yield of $H_2O$ has a high sensitivity to the OH concentration and give evidence that a minimum OH concentration is required for an effective $H_2O$ recycling.

The $\gamma_{H_2}$ shows an anti-correlated behavior to that of the $H_2O$ yield, however, as an exception, doubling of OH shows a lower yield than the reference in the mesosphere.





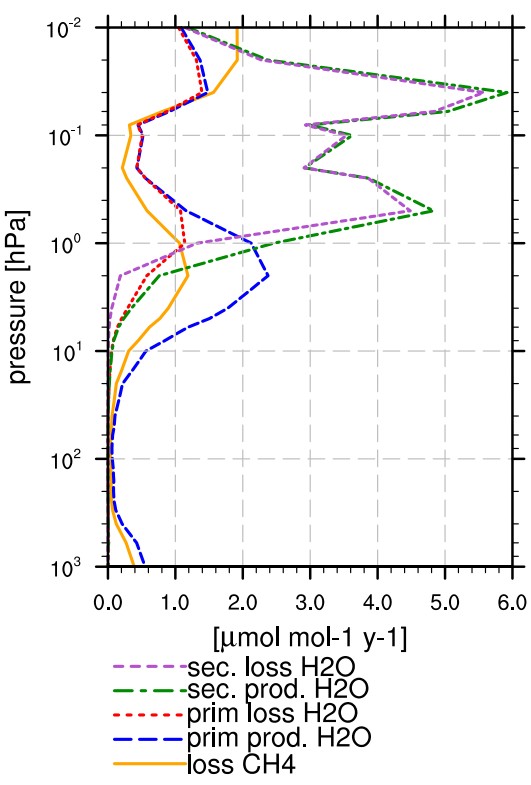

**Figure 6.** Separate loss of $CH_4$ and primary and secondary loss and production of $H_2O$ from box model simulation Ref.

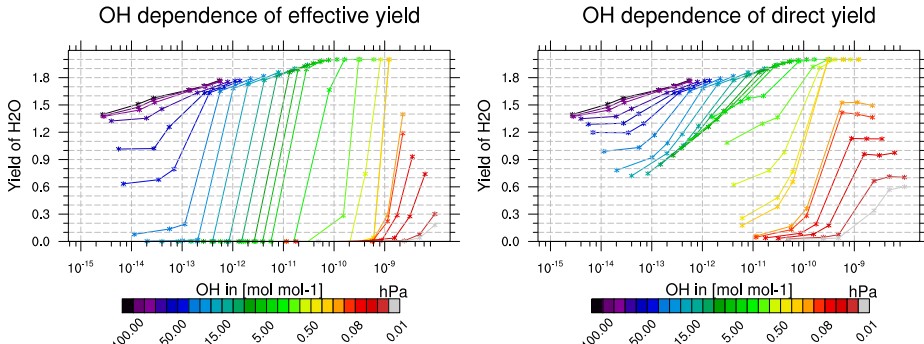

**Figure 7.** Effective yield (left) and direct yield (right) versus OH; colors indicate pressure level from low to high pressure.



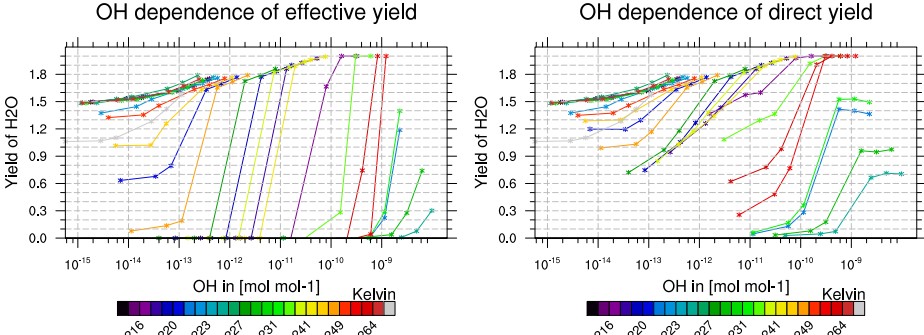

**Figure 8.** Effective yield (left) and direct yield (right) versus OH; colors indicate temperature from low to high temperature.

### 3.1.3 Dependencies on pressure and temperature

The results shown in the previous subsection indicate that there is an OH dependence in both the effective and direct yield. To investigate whether this dependency is systematic, simulated $H_2O$ yields are plotted as $\gamma_{H_2O}$ versus OH mixing ratio in Fig. 7. Generally, there is no linear correlation between these two parameters. However, a systematic dependence is evident for each

box, i.e. at each pressure level. The slope of the correlation is thereby dependent on the pressure level. For higher pressure the gradient is low and becomes steeper for lower pressure levels.

The slope of the correlation of OH and the direct yield (see Fig. 7 (right)) is smaller for pressure levels at 2–80 hPa than the slope of the effective yield (see Fig. 7 (left)) at corresponding pressure levels. Moreover, the effective yield has a sharp transition from low to high OH values, while the direct yield increases more gradually.

The scatter plots give evidence that in a certain range of pressure levels the yields exhibit a saturation-like behavior with respect to OH concentrations. Furthermore, there is no indication of a connection between the yield-OH-dependence and the temperature (see Fig. 8 and the non-ordered colors indicating the temperature), despite the fact, that reaction rates in the $CH_4 \rightarrow H_2/H_2O$-cycle are usually stronger impacted by temperature than by pressure.

We carried out additional sensitivity studies in order to investigate the temperature dependence of the yield on a given

pressure level. Results are displayed in Fig. 9. The simulation set-ups are identical to that of experiment Ref, except that temperature in every box was varied within -15 K to +15 K with 5 K steps. This temperature range is chosen as it represents a range exceeding day-night differences (less than $\pm5$ K) and the annual cycle (less than $\pm10$ K) in the tropics. In the lower stratosphere there is no indication of a significant temperature sensitivity of the effective and direct yields. The latter also does not show any significant sensitivity at higher altitudes. The effective yield in the upper stratosphere and mesosphere shows a

small dependence in a way that lower temperatures increase the yield and vice versa.

Consideration of the obvious vertical dependence and the very low temperature dependence gives evidence that not the physical parameters (temperature and pressure) itself are crucial for the $H_2O$ yield, but rather the chemical composition of the box. This chemical composition, however, changes with altitude (hence with pressure) and depends additionally on transport.





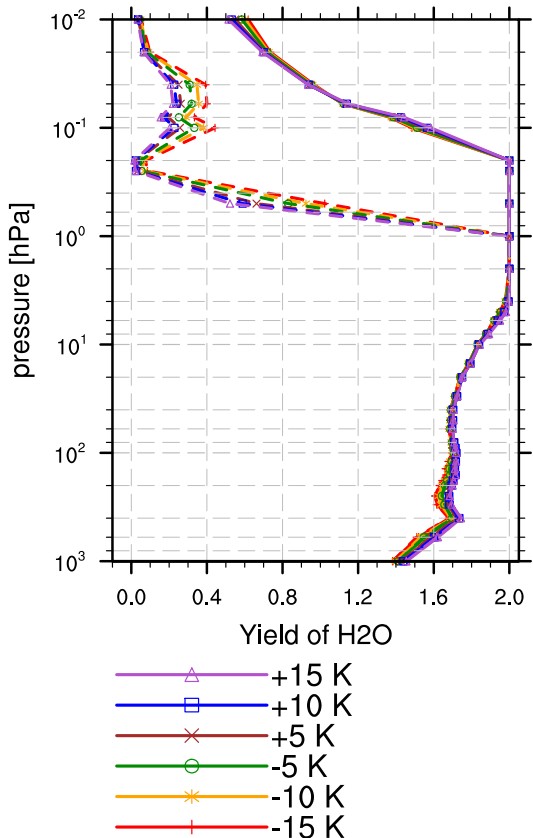

**Figure 9.** The pseudo vertical profile of the $H_2O$ yield calculated by the box model approach. Solid lines represent the direct yield, the dashed lines the effective yield and circles indicate the pressure levels of the model boxes. OH is kept constant to the initial values of the respective vertical box. The plot shows sensitivities concerning temperature. Temperature is varied from the standard atmosphere value by -15K (red), -10K (orange), -5K (green), +5K (brown), +10K (blue) and +15 K (purple).

## 3.2 Global model approach

As stated before, the box model approach does not take into account vertical transport and requires certain assumptions. Consequently, the boxes do not fully represent atmospheric conditions. To investigate the production of SWV in a comprehensive set-up, MECCA-TAG is applied in a global simulation with EMAC. The full chemistry of MECCA plus MECCA-TAG, which more than triples the amount of simulated tracers, increases the computational demands substantially. The additional tracers in the model defined by MECCA-TAG are basically counterparts of the tracers of the regular chemical mechanism and are marked (tagged) to be distinguishable from each other. In the following, these tracers are indicated by the label *tagged*. A spin-up simulation of 6 years with a reduced vertical resolution is carried out to pre-adjust tagged tracers. The results shown here originate from a subsequent simulation, which is executed for another two years model time.





Although the global simulation provides a three dimensional field, we focus in the current study on the vertical zonal mean profile of the yield of $H_2O$ from $CH_4$ oxidation ($\gamma_{H_2O}$) averaged over the tropics. An analysis of the zonal mean without meridional averaging (see Supplement Fig. S1) shows that the conclusions presented in this section also apply to a certain degree at mid latitudes. In the polar regions the analysis of the calculated yield is not useful as long periods without sunlight

and hence photolysis introduce substantial numerical errors into the calculation of $\gamma_{H_2O}$.

Figure 10 shows the vertical profile of the direct and effective yield of $H_2O$ in the tropics (23° S–23° N). Both match the vertical profile of the results of the box model simulation Exp1 superficially. However, there are certain differences.

First, the yield of $H_2O$ from $CH_4$ oxidation increases in the upper stratosphere and lower mesosphere to a value above 2, because the global model, unlike the box model, includes transport. The tagged intermediates (e.g. tagged $CH_3$, HCHO

etc.) which are produced at lower levels are transported upward and are finally converted to $H_2O$. This results in a production of more than two $H_2O$ molecules per oxidized $CH_4$ in one specific layer, because the additional production via transported intermediates is counted as well. In layers, where this increased production takes place, high OH concentration supports the conversion of the intermediates towards $H_2O$, since OH is the main driver of the chemistry.

The three topmost model layers in the upper mesosphere (0.06–0.01 hPa) are possibly subject to artifacts due to the nearby

top of the global model and are therefore not considered in this analysis. It is assumed that the trend, which is evident below 0.1 hPa, showing decreasing $\gamma_{H_2O}$ values also applies to the upper mesosphere, which would be similar to the box model results in the section above.

In Fig. 11 it also becomes obvious that the loss of $H_2O$ increases at higher altitudes. Additionally, the recycling of $H_2O$ contributes considerably to the effective yield. The photooxidation of $H_2O$ drives the continuously recycling of $H_2O$ to $H_2$ and

back, shifting the equilibrium between these two gases towards $H_2$.

Altogether, the separated $H_2O$ and $H_2$ loss/production terms of the global model are consistent with the box model findings. They also show a local maximum in loss of $CH_4$ and primary production of $H_2O$ below the stratopause and the strongly pronounced secondary loss and production of $H_2O$ in the middle and towards the upper mesosphere.

### 3.3 Ratio of H:$H_2$:$H_2O$

A different approach than the first two presented ones to determine $\gamma_{H_2O}$ in the stratosphere is to use the fact that the vertical profile of the H content in terms of atoms is fairly constant above the tropopause (see Fig. 12) compared to tropospheric variations. The H content in the stratosphere consists mostly of $CH_4$, $H_2O$, $H_2$, and, in the topmost layers, H. Other H carrying substances, such as OH, $HNO_3$, can be neglected for the H budget. The chemical regime determines the proportion between H, $H_2$ and $H_2O$, but the total H content is preserved.

The effective yield of $H_2O$ from $CH_4$ oxidation, as explained in previous sections, describes the net production of $H_2O$. Precisely it is an indicator for the interaction of loss and production of $H_2O$, further influencing the production of $H_2$ and H as well. As a first assumption, additional H from $CH_4$ oxidation should be partitioned to the reservoirs of H, $H_2$ and $H_2O$ in the same proportion as is present in the steady state. This is based on the supposition that it does not matter, whether the H, which is injected to the hydrogen cycling and reaches the indicated H reservoirs, comes from $CH_4$ or any other hydrogen





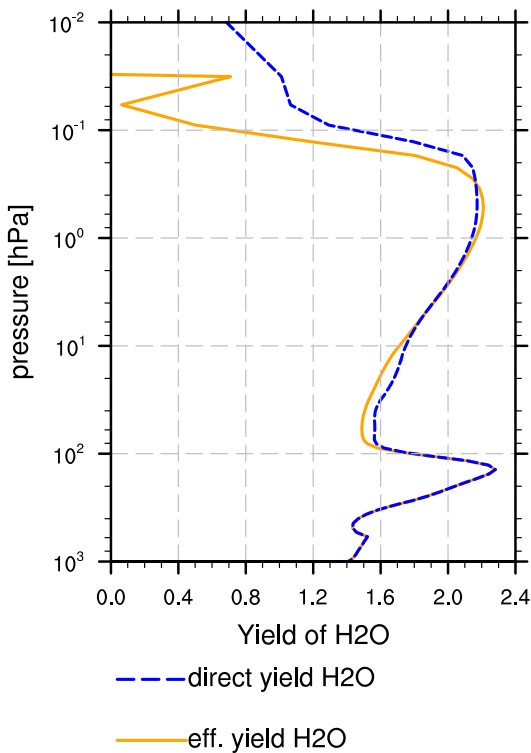

**Figure 10.** Effective and direct yield calculated from results of the global simulation in the tropics (23° S-23° N)

supply. If we assume further that the simulated proportion of H, $H_2$ and $H_2O$ at a certain level is approximately constant in time and that $CH_4$ is at higher layers the only additional hydrogen supply, we can determine the effective yield of $H_2O$ by $CH_4$ oxidation through the proportion of H atoms in $H_2O$ to the total hydrogen content of H, $H_2$ and $H_2O$. This proportion of the total hydrological content is subsequently called the H portion of $H_2O$.

5   In Fig. 13 the H portion of tagged and total $H_2O$ is plotted with respect to the sum of tagged and total H in the $CH_4$ oxidation products H, $H_2$ and $H_2O$, from the global experiment Exp2.

The H portion of $H_2O$ in the hydrogen budget is 2 in the troposphere and decreases to a minimum right above the tropopause. The hydrological cycle is producing a generally humid troposphere. Therefore, $H_2O$ in the lower layers of the atmosphere is prevailing versus $H_2$ and H, which are quickly oxidized as soon as they are produced. The minimum of the H portion of $H_2O$

10   above the troposphere can be explained by the freeze drying at the cold point. This reduces the H portion of $H_2O$ versus the one of H and $H_2$.

This minimum is not equally plain in the tagged $H_2O$. Note that tagged $H_2O$ in the troposphere is already lower than the total $H_2O$, since it is solely produced by $CH_4$ oxidation. When $CH_4$ ascends from the troposphere through the cold point into the stratosphere it continuously produces $H_2O$, although at low rates (due to low temperatures). Therefore, tagged $H_2O$ is still



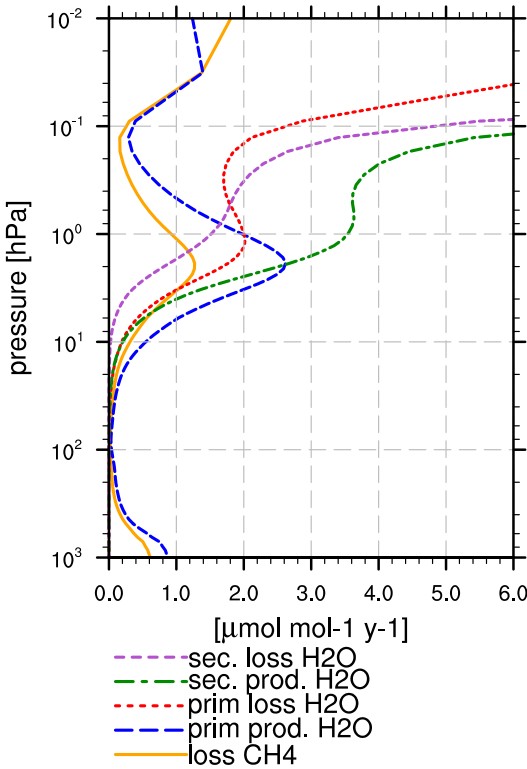

**Figure 11.** Separate loss of $CH_4$ and primary and secondary loss and production of $H_2O$ from the global simulation (23° S-23° N).

produced by $CH_4$ and even though it partly freezes out, the proportion to H and $H_2$ is not much impacted. However, in the lower stratosphere the mixing ratio of tagged $H_2$ increases, while $H_2O$ is still restrained by the cold point. This behavior becomes more apparent in case of the tagged species, since their absolute amounts are fairly low compared to the total ones.

Nevertheless, the H portion of tagged $H_2O$ and total $H_2O$ behave similar above the minimum at the tropopause, as seen

5 in the maximum around the stratopause and in the lower mesosphere and the strong decrease in the middle mesosphere and above. The general behavior of the vertical profile also agrees well with the above findings of the yield calculations using box model and global model results.

## 4  Discussion

The presented results show three different approaches in estimating $\gamma_{H_2O}$. Taking the results of the separate approaches together

10 gives the opportunity to discuss certain processes, which are differently parameterized and decisive for the yield estimation. We first want to discuss the general benefits and limitations of the approaches.





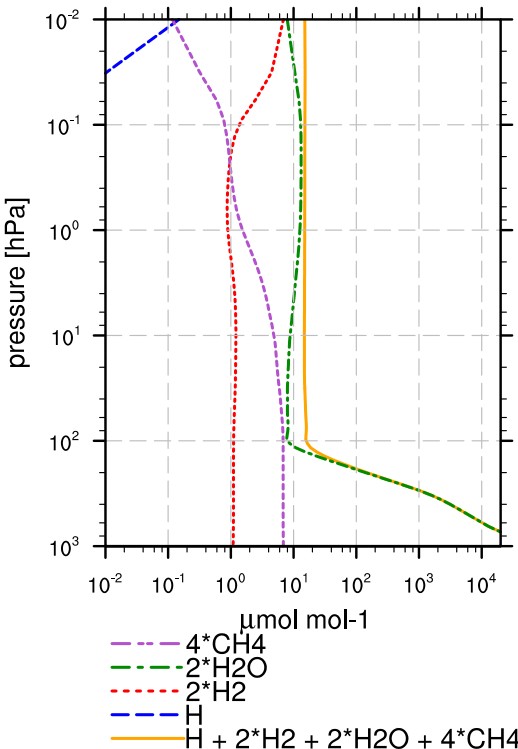

**Figure 12.** Annual zonal average of H content by species (in ppmv) over the tropics (23° S-23° N).

In the box model we have the opportunity to study a chemical regime without transport. It enables us to solely assess the involved chemical kinetics. Clearly, the box model chemistry does not fully represent the intended atmospheric conditions. Setting certain species to a constant value does change the chemical regime. However, without constraints on the chemical species the model would run into a new equilibrium, which changes the regime as well. It therefore needs careful weighing to specify, which species should be kept constant and which species should be allowed to re-adjust, to be able to simulate a representative chemical regime.

In the global model, we are not restricted to one vertical profile, but can evaluate the yield in three dimensions. Nevertheless, the effects of transport and chemical regime onto the yield cannot be separated, since transport influences the chemical regime. The vertical profile of $\gamma_{H_2O}$ is for this reason susceptible to changes in dynamical processes as for example the Brewer-Dobson circulation.

The third approach, which used the total H budgets and portions, helps to quantitatively evaluate the methods, which are calculating the effective yield. It shows the actual portion of hydrogen from $CH_4$ in the total hydrogen without a production and loss term, which is sensitive to variations in the chemical regime. Yet, this approach is not directly linked to the loss of



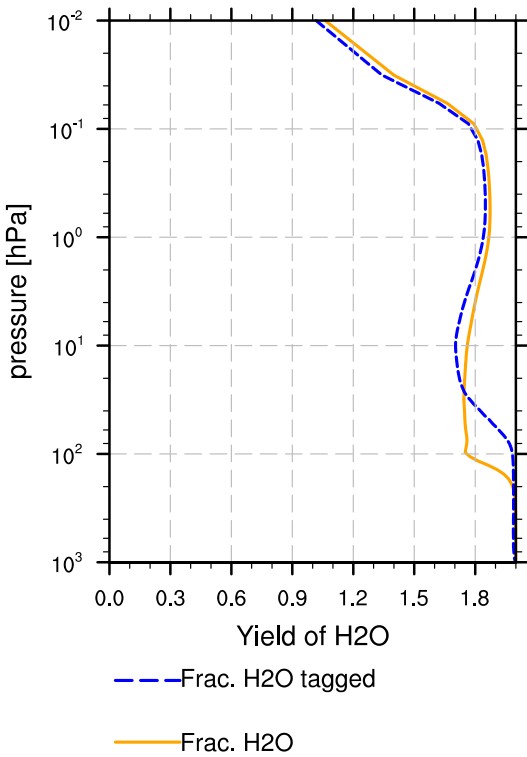

**Figure 13.** H portion of tagged and total $H_2O$ with respect to the tagged and total hydrogen content (H+2×$H_2$+2×$H_2O$), respectively.

$CH_4$ and it is not possible to explicitly resolve the influence of chemistry, since, for example, it is not clear if the decreasing values of $\gamma_{H_2O}$ in the mesosphere are due to the increasing loss of $H_2O$ or due to the reduced oxidation of $CH_4$.

Figure 14 shows the vertical profiles of the $H_2O$ yields and H portions calculated by the approaches described in the previous sections combined in one plot.

5  Comparing the results of the box model and the global model in the lower stratosphere, $\gamma_{H_2O}$ in the global model is lower than in the box model. This suggests that $CH_4$-produced $H_2O$ is transported into the stratosphere, where it is destroyed, adding to the loss of $H_2O$. This reduces $\gamma_{H_2O}$ while the oxidation of $CH_4$ is low, due to the exceptionally long lifetime of $CH_4$ due to low temperatures and low OH concentrations. In the upper stratosphere, global model $\gamma_{H_2O}$ is larger than box model $\gamma_{H_2O}$ and, more importantly, larger than 2, which is attributed to transport. This time, $CH_4$-derived intermediates are elevated and

10  produce $H_2O$ independent of the $CH_4$ oxidized in this region. This contradicts the assumption that two $H_2O$ molecules are immediately produced from $CH_4$ oxidation, since intermediates do play an important role.

In the middle mesosphere, box model and global model $\gamma_{H_2O}$ decrease substantially. Although, the topmost layers must be considered with caution due to potential artifacts, it is possible that the yield of the global model reaches values below zero. In the global model tagged $H_2O$ is transported into the mesosphere, where it is destroyed, due to the enhanced sink of $H_2O$ through





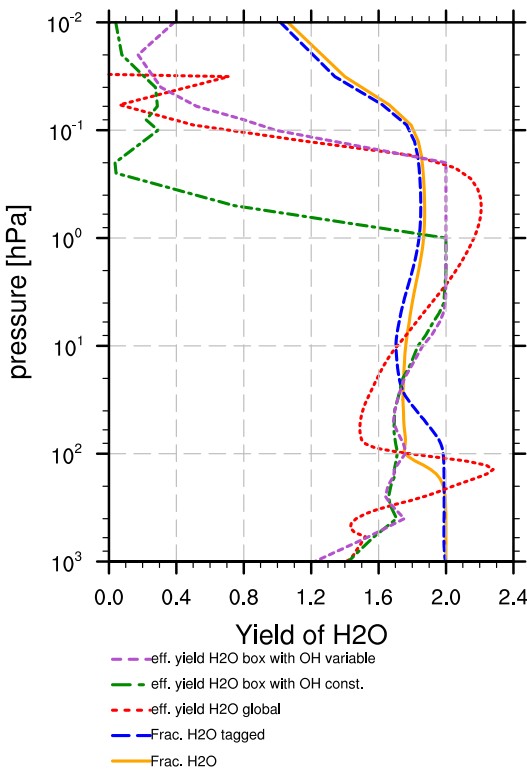

**Figure 14.** Comparison of all approaches determining the $H_2O$ yield: Effective yield by box model simulations with variable OH (purple, dashed) and fixed OH (green, dash-dotted), effective yield by global model simulations (red, dotted), H portion of total (yellow, solid) and tagged (blue, long dashed) $H_2O$ with respect to the hydrogen content.

photooxidation. The effective yield decreases below zero, since the loss of $H_2O$ becomes larger than the production of $H_2O$ ($\mathbf{P}^{I}_{H_2O} + \mathbf{P}^{II}_{H_2O} < \mathbf{L}^{I}_{H_2O} + \mathbf{L}^{II}_{H_2O}$). This emphasizes the importance of $H_2O$ destruction at higher altitudes, which particularly is not included, when parameterizing the chemical $\gamma_{H_2O}$ of $H_2O$ with two $H_2O$ molecules per $CH_4$ molecule oxidized.

Moreover, the effective yield in the box model setup with fixed OH profile drops down at $1\,\mathrm{hPa}$, while the yield of the box model with variable OH, (Exp1) and the global model (Exp2) do not drop until $0.2\,\mathrm{hPa}$. Additionally, Exp1 and Exp2 agree well concerning the altitude of the drop (the peak in Exp2 (red line) is most likely an artifact as discussed in Section 3.2). This suggests further that the chemical regime of the box model presented by the annual mean of the reference simulation (Ref) is not consistent with the chemical regime at the corresponding altitude concerning OH. The initialized and fixed value of OH at these levels is too low to realistically capture the chemical situation. This also shows that unconstrained OH is crucial and that the vertical profile of OH of simulation Exp1 in this region better agrees with the OH in the global simulation Exp2.

The H portion of $H_2O$ in the hydrogen content matches qualitatively the results of the yield calculations in the box and global model approach. MECCA-TAG again enables us to focus on H in $H_2O$ particularly from $CH_4$ oxidation and to ignore the H





from other sources. The minimum of the H portion of $H_2O$ in the lower stratosphere and its maximum close to the stratopause and in the lower mesosphere therefore shows that the production of $H_2O$ from $CH_4$ oxidation relative to the production of $H_2$ from $CH_4$ oxidation is smaller in the lower stratosphere and becomes larger towards the upper stratosphere. Accordingly, we conclude that our estimation that $\gamma_{H_2O}$ differs significantly from 2 in the lower stratosphere is reliable.

Altogether, the different approaches yield consistent results. All suggest a yield of less than 2 in the lower stratosphere, varying between 1.5 and 1.7. The smallest value is estimated in the global simulation Exp2, where the yield is larger than the one of le Texier et al. (1988), which is $\gamma_{H_2O}$=1.3 at corresponding altitudes. The results of le Texier et al. (1988) also showed a maximum around 1 hPa, which is consistent with our results, albeit being a bit above 1.8 and with that lower than our estimate of 2 (or more in case of the global simulation) in that region.

Overall, the estimated yield of $H_2$ from le Texier et al. (1988) and the yield of $H_2$ estimated by the box model approach are consistent as well. While our resulting $\gamma_{H_2O}$ is larger than in le Texier et al. (1988), the $\gamma_{H_2}$ is lower. Still, the vertical profiles of $\gamma_{H_2}$ in both studies are comparable.

The fundamental study of le Texier et al. (1988) does not capture the influence of the increasing loss of $H_2O$ at higher altitudes. They only considered the direct yield of $H_2O$ and do not include $H_2O$ loss in their calculation. Nevertheless, the
findings in our study show, that the difference between effective and direct yield becomes only apparent above 0.1 hPa and le Texier et al. (1988) do not discuss results above this pressure level. Yet, we see it critical to use the results of le Texier et al. (1988) to justify the approximation of $\gamma_{H_2O}$=2 at lower altitudes.

Hurst et al. (1999) calculated a net production of $H_2O$ of $1.973 \pm 0.003$, which includes a loss of H via $H_2$ of $0.027 \pm 0.003$. These values differ from our findings in the box model approach. Our estimated $\gamma_{H_2O}$ is smaller and our $\gamma_{H_2}$ is larger than
estimated by Hurst et al. (1999). As noted before, by using observational data it is not possible to distinguish between $H_2$ from the troposphere and $H_2$ produced by H from $CH_4$, which results in this rather low net production of $H_2$. Assume, for example, that $H_2$ is not produced in the stratosphere. The mixing ratio of $H_2$ will then decrease with respect to altitude. However, the contribution from $CH_4$ oxidation onto $H_2$ fills up the oxidized molecules, and only if $\gamma_{H_2} \cdot [CH_4]$ is larger than the total loss of $H_2$, observed $H_2$ and $CH_4$ are anti-correlated. Using the kinetic tagging gives us the opportunity to distinguish between the
total loss of $H_2$ and the loss of those $H_2$ molecules carrying H from $CH_4$. Our findings provide therefore an additional insight into processes, which determine the observed vertical profiles and provide estimates for the contribution of $CH_4$ separated from the background $H_2$ and $H_2O$.

Summarizing, our results suggest that applying $\gamma_{H_2O}$=2 as the contribution to $H_2O$ by the oxidation of $CH_4$ in climate models likely overestimates the kinetic yield of $H_2O$ in the lower stratosphere and in the mesosphere above 0.2 hPa. We admit,
however, that a small fraction of $H_2O$ should also be produced from $H_2$ ascending from the troposphere. This likely reduces the SWV bias in GCMs simulations using the approximation of $\gamma_{H_2O}$=2, since those models do not include a separate $H_2O$ production from $H_2$ oxidation. Nevertheless, to be punctilious, the yield of $H_2O$ from $CH_4$ oxidation should be distinguished from the net chemical production of $H_2O$. In subsequent studies, we intend to apply the tagging method for estimating a $\gamma_{H_2O}$ from $H_2$ oxidation ($\gamma_{H_2O}(H_2)$). $H_2$ and $CH_4$ may oxidize at a similar rate, but the resulting products are different, which likely
results in a varied $\gamma_{H_2O}$ with respect to the source gas (i.e. $\gamma_{H_2O}(CH_4) \neq \gamma_{H_2O}(H_2)$).





Another important disadvantage of the parameterization as in Eq. (1) with $\gamma_{H_2O}$=2 is that it does not account for the loss of $H_2O$ in the mesosphere. Even though $CH_4$ oxidation becomes negligible at these altitudes, this simple parameterization does not consider that $H_2O$ gets chemically destroyed. Strictly speaking, the loss of $H_2O$ is independent of $CH_4$ and should potentially be included separately. MacKenzie and Harwood (2004) and McCormack et al. (2008) presented, for example,

sophisticated parameterizations, which target this issue in their 2D atmospheric models. Based on our results, we recommend to apply a parameterization, which is not solely based on the loss of $CH_4$, but accounts for the reduced yield in the lower stratosphere and also includes the loss of $H_2O$.

Besides this, transport of intermediates is an important factor for the vertical profile of the $\gamma_{H_2O}$. It must be noted that atmospheric transport is not constant in time. The Brewer-Dobson circulation for example is predicted to change in future

climate projections (Butchart et al., 2010). For a comprehensive parameterization of $\gamma_{H_2O}$ these changes in transport must be taken into account. However, changes in transport depend on various factors and are therefore difficult to be included into $\gamma_{H_2O}$ parameterizations. This raises the question, whether a simplified parameterization of $\gamma_{H_2O}$ is indeed applicable for future climate projections or if it is necessary to simulate the full-chemistry, if an accurate SWV is desired.

## 5 Conclusions

In this study, we present a comprehensive evaluation of current assumptions and estimates of the chemical yield of $H_2O$ from $CH_4$ oxidation in the middle atmosphere. We show results of three different approaches to estimate $\gamma_{H_2O}$ and discuss certain advantages and challenges.

We conclude that the widely used assumption that one $CH_4$ molecule produces two water molecules overestimates the kinetic $H_2O$ production in the stratosphere up to 4 hPa and in the mesosphere above 0.2 hPa. Our results show that a local

yield larger than 2 in certain areas is possible through ascended intermediates. In addition to that, transport is generally an issue when dealing with kinetic yields, since it influences the chemical regimes at all altitudes. It also makes the interpretation of the presented approaches challenging, when these are investigated separately.

Nevertheless, the separate approaches presented in this study, show consistently that $\gamma_{H_2O}$ is substantially lower than 2 in the lower stratosphere, has a local maximum between 0.2 and 0.4 hPa and is exceedingly low in the upper mesosphere. We find

a low $\gamma_{H_2O}$ in the middle and upper mesosphere, since the loss of $H_2O$ at higher altitudes increases, shifting the equilibrium between $H_2O$ and $H_2$ towards $H_2$. The chemical loss is therefore a crucial factor for the correct parameterization of SWV production from $CH_4$ oxidation. At some point, the loss of $H_2O$ is so strong that $H_2O$ is effectively destroyed per oxidized $CH_4$.

An additional result from the box model simulation is that the chemical yield of $H_2O$ depends on the OH concentration and

more general on the chemical kinetics. A strong temperature dependence, however, could not be detected.

Furthermore, the presented results agree with earlier kinetic estimates of $\gamma_{H_2O}$ from le Texier et al. (1988), who state that not exactly two molecules are produced from $CH_4$ oxidation. Furthermore, our results give an additional insight into observations (e.g. Hurst et al. (1999); Rahn et al. (2003)), which are limited in detecting the chemical origin of $H_2O$.




Overall, the results of the separate approaches give evidence that calculating the yield of $H_2O$ from $CH_4$ oxidation requires the loss of $H_2O$ to be taken into account, making the task of creating a simple parameterization challenging. The latter also requires to admit a critical amount of assumptions about uncertain factors for an adequate atmospheric simulation. We therefore recommend, in order to maintain as much certainty as possible concerning the chemical yield of $H_2O$, to implement a simplified

$H_2O$ chemistry including the most important reactions determining the $H_2O$ yield. The extent of the resulting subset of the chemical mechanism is determinative for the correct representation of the $H_2O$ content in the middle atmosphere. However, it must be noted that a set of reactions required for the comprehensive simulation of $H_2O$ kinetics is not substantially different from the one incorporated in the full chemistry setup and is therefore less beneficial in terms of computational resources than a parameterized model. Nevertheless, as stated before, a too simple parameterization introduces uncertainties, which makes

it challenging to preserve the required accuracy for applications in the simulation of climate projections, where atmospheric dynamics (e.g. the Brewer-Dobson circulation) and chemistry potentially differ from the present-day atmosphere.

The investigations presented in this study should serve as a basis for future studies concerning the chemical yield of $H_2O$ in the stratosphere and mesosphere. The gained knowledge can be used to derive new parameterizations of the chemical yield of $H_2O$ for a potential application in GCMs.

*Code and data availability.* The Modular Earth Submodel System (MESSy) is continuously developed and applied by a consortium of institutions. The usage of MESSy and access to the source code is licensed to all affiliates of institutions, which are members of the MESSy Consortium. Institutions can become a member of the MESSy Consortium by signing the MESSy Memorandum of Understanding. More information can be found on the MESSy Consortium Web-site (http://www.messy-interface.org). The data of the box model simulations described above is available in the supplement. Data of the global simulation is available upon request from the corresponding author.

*Competing interests.* The authors declare that they have no conflict of interest.

*Acknowledgements.* We acknowledge the DLR internal project KliSAW (Klimarelevanz von atmosphärischen Spurengasen, Aerosolen und Wolken), which provided the financial basis for the presented study.

The model simulations have been performed at the German Climate Computing Centre (DKRZ) through support from the Bundesministerium für Bildung und Forschung (BMBF).

We used the NCAR Command Language (NCL) for data analysis and to create some of the figures of this study. NCL is developed by UCAR/NCAR/CISL/TDD and available on-line: http://dx.doi.org/10.5065/D6WD3XH5.

We furthermore thank all contributors of the project ESCiMo (Earth System Chemistry integrated Modelling), which provides the reference profiles and initial conditions as well as Christoph Kiemle for his internal review and valuable comments on the manuscript.





**Supplement**

– meccanism.pdf : The applied chemical mechanism of the box model and EMAC simulations.

– supplement.pdf : Including 2D Profiles of the EMAC simulations in terms of $\gamma_{H_2O}$ and the ratio of H:$H_2$:$H_2O$, as well as the data of the box model simulations.





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

<