# Peer review of "Investigating the yield of H2O and H2 from methane oxidation in the stratosphere"

_Atmospheric Chemistry and Physics, 2018_

## Referee Comment (RC1) · Anonymous Referee #1 · 23 Mar 2018

This is an interesting study which is certainly appropriate for publication in ACP. As the title says, the primary motivation is to investigate the oxidation of methane in the stratosphere, to determine how much water vapor and H2 is produced, and to the sensitivity of this production to certain geophysical parameters.

My most serious concern is that this study fails to provide any discussion and comparison to the obviously relevant study by Wrotny et al. ("Total hydrogen budget of the equatorial upper stratosphere"; JGR 2010). Some appropriate discussion should therefore be added. While it seems clear that this manuscript will disagree with some of the high yield values found in the Wrotny study, the upper stratosphere/lower mesosphere does appear to be the one region where this study shows a yield greater than 2 (Figure 10).

[Figure]

Page 5 line 12 – "The equator is chosen for its negligible seasonal cycle." While the equator is a reasonable choice because some seasonal cycles are smaller, the change in H2O entering the stratosphere at the equator is, among other things, certainly not "negligible".

Page 5 line 29 – "is not known" should be "are not known"

Page 7 line 6 – "once" should be "at a time".

Figure 5 – The order of the lines in the legend is a bit strange and confusing, being neither high to low nor low to high OH. Please make this easier for the reader.

Page 13 line 3 – This short summary paragraph is confusingly written, especially given the use of the phrase "on the other hand". Unless I'm missing something, increasing OH concentration simply increases the yield of H2O by both the direct and effective measures with the difference between direct and effective being largest at the highest altitudes.

Figure 6 – Perhaps I am missing some important point, but it seems to me that this figure and the accompanying text on page 12 is in the section "Sensitivity with respect to OH". Wouldn't it be much more appropriately placed right after the introduction of equation (3), which forms the basis of the terms being plotted?

Figure 12 – This is an extremely important figure, yet it is plotted on a log scale which makes it difficult to quantitatively determine many of the values of interest. The species could all be put on the same scale with appropriate offsets and multipliers. In particular, it would be interesting to see the H2 variation with altitude in the stratosphere on a linear scale. It is not necessary to show the decrease in water vapor with increasing altitude in the troposphere, so this figure could certainly be started at 100 hPa.

Page 21 line 1 – "explicitely" should be "explicitly".

Page 23 line 16 – "Yet, we see it critical to use the results of le Texier et al. (1988) to justify the approximation of H2O=2 at lower altitudes." I don't understand this sentence.

Figure S2 – This figure would be much more informative if the colors were not all red. It seems to me that the color scale could be run from ∼1 to just over 2.

[Figure]

---

## Referee Comment (RC2) · Anonymous Referee #2 · 26 Mar 2018

Frank et al. present a thorough investigation into stratospheric methane oxidation, and challenge a long-hold assumption in GCMs without online chemistry, namely that one molecule of CH4 produces exactly two molecules of H2O. Using three separate methods, they show that this assumption is incorrect over many altitudes. I find the study design well thought through and well presented, and suitable for publication in ACP.

One I idea I had around improving the manuscript, is that the authors encourage the use of comprehensive parametrizations in GCMs – but I am not clear on what exactly these parametrizations should be? I wondered if it could be worth adding a subsection near the end called 'recommendations for GCMs without online chemistry,' or similar.

I'm also aware of more recent parametrizations for methane oxidation, e.g. as discussed in Oman et al. (2008), whereby the rate of methane oxidation takes into account pressure, latitude and age-of-air. I think that discussing these more recent parametrizations would round out the discussion nicely.

Minor comments: - The paper will benefit from copy editing for English, and the authors might want to find a native English speaker or two to help with that when they resubmit the final version.

- P1L18: how about at polar latitudes?

- P3L20: I guess you mean it's not vertically well mixed, c.f. zonally.

- P3L25: please state what sort of model you use in (3).

- P3L27 and elsewhere: note the correct spelling of 'explicitly.'

- P3L28-30: it wasn't clear to me what you mean by this sentence.

- P4L24: can you list the simulated intermediates here (if practical)?

- P4L30: please define acronyms, e.g. NMHCs, HCFCs

- P4L30: how do you define HOx? H+OH+HO2?

- P5: Please also define NOx, ClOx and BrOx

- P5L28: I'd like to know more about your RC1SD-base-10 EMAC simulation. E.g., from the name, can it be inferred that dynamics are specified to a reanalysis?

- Fig.2: where is H2O being lost to in the mesosphere?

- Table 2: the authors might want to consider adding an extra column stating whether the simulation is a box model or CCM simulation.

- P15L21-23: Can you comment on what in particular is important regarding chemical composition of the box?

- P24L13: I think it's now fairly well recognised that online chemistry is necessary in

many respects, e.g. the Southern Hemisphere circulation response to CO2 via O3 changes (Chiodo and Polvani, 2017).

References: Chiodo, G., and L. M. Polvani (2017), Reduced Southern Hemispheric circulation response to quadrupled CO2 due to stratospheric ozone feedback, Geophys. Res. Lett., 44, 465–474, doi:10.1002/2016GL071011.

Oman, L., Waugh, D.W., Pawson, S., Stolarski, R. and Nielsen, J. E., Understanding the changes of stratospheric water vapour in coupled chemistry-climate model simulations, J. Atmos. Sci., 65, 3278-3291, 10.1175/2008JAS2696.1, 2008.

---

## Referee Comment (RC3) · Anonymous Referee #3 · 28 Mar 2018

This paper purports to test the assumption that the yield of water vapor from methane oxidation is equal to 2. I found the paper to be confusingly written and likely to mislead readers who do not know the field well.

The most important thing: reading this paper might lead the reader to conclude that the assumption that dCH4/dH2O = 2 is not a good one. In fact, we have many observations (they are referenced in this paper) that show it is an excellent assumption throughout most of the stratosphere. I agree that the assumption breaks down at high altitudes.

The reason the assumption is good in the lower stratosphere, even though the calculated yield there is less than 2, is that the lifetime of CH4 there is very long (100 years). Almost all of the oxidation of methane in the stratosphere is occurring in the mid-stratosphere, where the yield is 2. This air is transported down into the lower

stratosphere, so the yield in the lower stratosphere just reflects mid-stratosphere photochemistry.

This needs to be clearly laid out in the paper. Otherwise, readers will be misinformed.

Assessing the quality of the assumption that dH2O/dCH4 = 2 would require a different analysis. All one would have to do is show regressions of H2O versus CH4 in various regions of the stratosphere (from either observations or models with full stratospheric chemistry). This comparison would show you if that assumption is good.

In fact, the paper is really about H photochemistry, not the assumption that dH2O/dCH4 = 2. There's a lot of discussion in the paper that revolves around the details of stratospheric photochemistry. So one possible suggestion that I think would improve the paper would be to remove the present motivation of the paper (testing if dH2O/dCH4 = 2) and replace it with a more accurate characterization of the work described (investigating H photochemistry and sensitivities).

A few smaller comments: 1) I would eliminate Fig. 1 below 100 hPa. This region is not relevant to the paper.

2) Why do the authors spend so much time looking at OH sensitivity? That section should be motivated better.

3) I don't understand why the direct and effective yields of water vapor in the lower stratosphere are equal. The direct yield is the water vapor produced directly from methane oxidation. However, there's also a contribution from oxidation of H2 (lifetimes of CH4 and H2 are similar in the lower strat.). That would be included in the effective yield. Thus, the effective yield should be larger than the direct yield, right? I'm confused.

4) This emphasizes that I don't particularly understand the way the authors have defined effective and direct yield. It seems to me that direct yield should be production of water directly from methane oxidation and effective yield should be the direct production plus the yield of water vapor from H2 oxidation and minus the loss of H2O from photochemistry. Is this how they view their definitions? If so, they should perhaps re-phrase that part of the manuscript.

---

## Author Comment (AC1) · 27 Apr 2018

**Answer to the referee # 1**

April 27, 2018

Dear referee,

we thank you for the thoughtful comments on our manuscript. In the following we reply to them point-by-point. The indicated pages of the answers relate to the discussion paper.

> (1) My most serious concern is that this study fails to provide any discussion and comparison to the obviously relevant study by Wrotny et al. (Total hydrogen budget of the equatorial upper stratosphere; JGR 2010). Some appropriate discussion should therefore be added. While it seems clear that this manuscript will disagree with some of the high yield values found in the Wrotny study, the upper stratosphere/lower mesosphere does appear to be the one region where this study shows a yield greater than 2 (Figure 10).

> We thank the referee for pointing us to this study, which certainly adds some valuable aspects to our discussion. Some fitting aspects are added to the introduction and the discussion of the revised manuscript.

Text added to the manuscript:

**page 2** By analysing satellite based measurements Wrotny et al. (2010) derived a production of $H_2O$ over loss of $CH_4$ ratio of 2.0–3.7 in the upper stratosphere between 1.0–4.6 hPa, which is clearly $\geq 2$.

**page 19** This sum (H + 2\*$H_2$ + 2\*$H_2O$ + 4\*$CH_4$) of 15 $\mu$mol mol$^{-1}$ is in accordance with the estimate derived with the CHEM2D model by Wrotny et al. (2010) for the sum of $H_2$ + $H_2O$ + 2\*$CH_4$ being $\sim 7.5$ $\mu$mol mol$^{-1}$ (i.e. one half of $\simeq 15$ $\mu$mol mol$^{-1}$). The individual abundances of $H_2$, $H_2O$ and $CH_4$ also agree well with each other.

**page 21** This is furthermore consistent with the findings of Wrotny et al. (2010), who calculated a yield larger than 2 in this area as well.

**page 23** The study of Wrotny et al. (2010), based on a correlation analysis of satellite measurements, derived a yield of 2.6–2.7 at 1.0 hPa (depending on the satellite product and error assumptions). These are larger than our estimate, which is less than 2.3. Nevertheless, we agree that the yield can be larger than 2, but a direct comparison of our model results with the measurement based derivation of Wrotny et al. (2010) is not possible for the arguments given above.

> (2) Page 5 line 12 - "The equator is chosen for its negligible seasonal cycle." While the equator is a reasonable choice because some seasonal cycles are smaller, the change in H2O entering the stratosphere at the equator is, among other things, certainly not negligible.

> We agree with the referee that there are seasonal changes in the $H_2O$ entering the stratosphere (i.e. tape recorder signal), these are taken into account in our set-up, since the MECCA-TAG distinguishes between the transported $H_2O$ and that produced by $CH_4$. However, our argument for the equator was indeed a bit sloppy. We chose the equator to avoid the polar night, where photochemistry is virtually inactive thoughout long parts of the year.
> We deleted the corresponding half sentence in the abstract.

**page 5:**

**Old:** The equator is chosen for its negligible seasonal cycle.

**New:** The equatorial region is chosen for mainly two reasons: (1) the equatorial region is in terms of photochemistry most active and (2) we avoid the inactive photochemistry during the polar night.

**Abstract:**

**Old:** We focus representatively on the tropical zone between 23° S-23° N, where seasonal variations are negligible.

**New:** We focus representatively on the tropical zone between 23° S-23° N.

> (3) Page 5 line 29 - is not known should be are not known

> Thank you for spotting this. We corrected it!

> (4) Page 7 line 6 - once should be at a time.

> We changed it as suggested to "at a time".

> (5) Figure 5 - The order of the lines in the legend is a bit strange and confusing, being neither high to low nor low to high OH. Please make this easier for the reader.

> The legend was indeed confusing. We reorderd the legend accordingly. We also rearranged the legend in Figure 9 in the same manner to save space.

> (6) Page 13 line 3 - This short summary paragraph is confusingly written, especially given the use of the phrase on the other hand. Unless Im missing something, increasing OH concentration simply increases the yield of H2O by both the direct and effective measures with the difference between direct and effective being largest at the highest altitudes.

> Yes, this is true. We also wanted to explain that the larger difference at higher altitudes is likely caused by the chemical regimes at these altitudes. We like the suggested order and reformulated the paragraph accordingly so that it is easier comprehensible.

**page 13:**

**Old:** Summarizing, reduction of the OH concentrations leads to a proportionally larger decrease in the $H_2O$ yield at higher altitudes owing to the differences in the chemical regimes. On the other hand, increasing the OH concentration also increases the direct and foremost the effective yield of $H_2O$.

**New:** Summarizing, increasing OH concentrations lead to higher direct and effective $H_2O$ yields. Both yields show when varying OH a larger difference to the reference at higher altitudes, indicating that the sensitivity of the chemical regime with respect to the OH concentration increases with altitude.

> (7) Figure 6 - Perhaps I am missing some important point, but it seems to me that this figure and the accompanying text on page 12 is in the section Sensitivity with respect to OH. Wouldnt it be much

more appropriately placed right after the introduction of equation (3), which forms the basis of the terms being plotted?

It is true that Figure 6 and the accompanying text is in the Section 3.1.2., where it does not exactly fit well. It is a very good suggestion to move it next to Equation (3).

(8) Figure 12 - This is an extremely important figure, yet it is plotted on a log scale which makes it difficult to quantitatively determine many of the values of interest. The species could all be put on the same scale with appropriate offsets and multipliers. In particular, it would be interesting to see the H2 variation with altitude in the stratosphere on a linear scale. It is not necessary to show the decrease in water vapor with increasing altitude in the troposphere, so this figure could certainly be started at 100 hPa.

Thank you for pointing this out. We changed the horizontal axis to a linear scale and shifted H by 3.0 ppmv for the sake of visibility. However, we prefer to keep the vertical axis down to 1000 hPa to be comparable with the other plots throughout the paper.

(9) Page 21 line 1 - explicitely should be explicitly.

That is corrected.

(10) Page 23 line 16 - Yet, we see it critical to use the results of le Texier et al. (1988) to justify the approximation of H2O=2 at lower altitudes. I dont understand this sentence.

We wanted to express that we (1) do not recommend to use the approximation of yield=2 at lower altitudes and (2) emphasize that this is, strictly speaking, also not supported by the results of le Texier et al. (1988) although often declared as the reference. We reformulated this sentence to be more precise.

**page 20:**

**Old:** Yet, we see it critical to use the results of le Texier et al. (1988) to justify the approximation of H2O=2 at lower altitudes.

**New:** Furthermore, le Texier et al. (1988) is often cited as the reference for the assumption of $\gamma_{H_2O}$=2. However, in the lower stratosphere our results and those of le Texier et al. (1988) actually agree that $\gamma_{H_2O}$ is less than two, which objects the assumption of a constant $\gamma_{H_2O}$=2.

(11) Figure S2 - This figure would be much more informative if the colors were not all red. It seems to me that the color scale could be run from ∼1 to just over 2.

We reduced the scale as suggested. Now some more features are apparent in Figure S2.

---

## Author Comment (AC2) · 27 Apr 2018

**Answer to the referee # 2**

April 27, 2018

Dear referee,
thank you for the positive review and valuable comments on our manuscript. We address them point-by-point below. The indicated pages of the answers relate to the discussion paper.

**Major comments**

> One I idea I had around improving the manuscript, is that the authors encourage the use of comprehensive parametrizations in GCMs – but I am not clear on what exactly these parametrizations should be? I wondered if it could be worth adding a subsection near the end called recommendations for GCMs without online chemistry, or similar.

> This is a very good idea. We moved some parts concerning this into a corresponding subsection at the end of the discussion and included some additional suggestions for parameterizations. Unfortunately, we can not be very specific, since we have not yet tested the suitability of our recommendations. This will certainly be addressed in subsequent studies.

**Added text on page 24:**
Nevertheless, one could start with a parameterization as introduced by Eq. (1), however, with a pressure dependent $\gamma_{H_2O}(p)$ derived from our vertical yield profiles. This adds a vertical dependency to the chemical production of $H_2O$ per $CH_4$ oxidized. As long as no large variations or trends in the stratospheric transport are expected within the simulation period, our profile is a good approximation. The limitation is, however, that the pressure dependence is likely to change with changing climate.

At higher altitudes (above 0.2 hPa) the yield in Eq. (1) could be replaced or supplemented by an explicit parameterization of the chemical loss of $H_2O$, mostly via photolysis and the reaction with $O(^1D)$, see MacKenzie and Harwood (2004) and McCormack et al. (2008). In the simplified methane chemistry of EMAC, for example, a predefined $O(^1D)$ is also used for the reaction with $CH_4$ and could be reused for the reaction with $H_2O$. Again, the same limitation holds: under climate change, water vapour, and photolysis rates are likely to change.

> Im also aware of more recent parametrizations for methane oxidation, e.g. as discussed in Oman et al. (2008), whereby the rate of methane oxidation takes into account pressure, latitude and age-of-air. I think that discussing these more recent parametrizations would round out the discussion nicely.

> Thank you for the note on the paper of Oman et al. (2008). Indeed, they use a paramterization (based on Austin et al (2007)) of the loss of $CH_4$ ($\frac{d}{dt}[CH_4]$). Note that we evaluate the yield of $H_2O$ per oxidized $CH_4$, i.e. $\gamma_{H_2O}$. Both are in the parameterization of the $H_2O$ production independent of each other:
>
> $$\frac{d}{dt}[H_2O] = -\gamma_{H_2O} \cdot \frac{d}{dt}[CH_4] \ .$$
>
> Austin et al. (2007) and Oman et al. (2008) still assume a yield of two $H_2O$ molecules per oxidized $CH_4$. For this reason, the study of Austin et al. (2007) is mentioned in the introduction of our manuscript. A similar parameterization is for example introduced by Monge-Sanz et al. (2013).

We are hesitating to expand the discussion of the yield towards the discussion of the parameterization of the loss of $CH_4$ ($\frac{d}{dt}[CH_4]$), since this can be chosen individually and independent of the assumptions concerning the yield.

Our concern is further that the mentioned parameterizations do not consider the loss of $H_2O$, which is indeed mentioned by Oman et al. (2007) as well. There are, however, rather recent parameterizations of MacKenzie and Harwood (2004) and McCormack et al. (2008), which include the loss of $H_2O$ applied in 2-D models and are already part of the discussion (Page 24, line 4).

Nevertheless, we added a reference to Oman et al. (2008) in the introduction, since it fits to the other mentioned papers (Monge-Sanz et al., 2013; ECMWF, 2007; Austin et al., 2007; Boville et al., 2001; Mote, 1995; Eichinger et al., 2015) using the assumption of a constant yield of 2.

**Minor comments**

The paper will benefit from copy editing for English, and the authors might want to find a native English speaker or two to help with that when they resubmit the final version.

Thank you for this suggestion. Please note that copy editing for language is in any case applied by the journals editorial office before final publication.

P1L18: how about at polar latitudes?

During the polar night, due to the virtual absence of OH, the method is not applicable. This is discussed in the paper. We added a sentence in the abstract as well.

**Abstract:**

**Old:** It is found in the global approach that presented results are mostly valid for mid latitudes as well.

**New:** It is found in the global approach that presented results are mostly valid for mid latitudes as well. During the polar night the method is not directly applicable.

P3L20: I guess you mean its not vertically well mixed, c.f. zonally.

Yes, indeed, the vertical mixing is of most concern here.

**page 3:**

**Old:** As an additional remark, it should be noted that difficulties with yield estimates can be expected especially in the stratosphere, as it is not as well mixed as the turbulent troposphere.

**New:** As an additional remark, it should be noted that difficulties with yield estimates can be expected especially in the stratosphere, as it is vertically not as well mixed as the turbulent troposphere.

P3L25: please state what sort of model you use in (3).

Thank you, that is indeed not clear here. We added a corresponding note.

**page 3:**

**Old:** For the third approach (3), we rely on the assumption that the hydrogen budget in the stratosphere is conserved, mostly consisting of fractions of H, H2, H2O and CH4.

**New:** For the third approach (3) we again use the model results of a global simulation with EMAC. This approach relies on the assumption that the hydrogen budget in the stratosphere is conserved, mostly consisting of fractions of H, H2, H2O and CH4.

P3L27 and elsewhere: note the correct spelling of explicitly.

Thank you for pointing this out. We corrected it throughout the manuscript.

P3L28-30: it wasnt clear to me what you mean by this sentence.

We want to state that the study of Johnston and Kinnison (1998) is an additional example for estimating the yield of methane oxidation. This yield is the impact of CH4 on ozone instead of H2O. We reformulated the sentence to be more concise.

**page 3:**

**Old:** Despite that this study focuses on the tropospheric and lower stratospheric ozone (O3), it is a practical example on the derivation of atmospheric trace gas yields.

**New:** The study of Johnston and Kinnison (1998) is an example for estimating a yield from CH4 oxidation, although it focuses on CH4 impacts on ozone (O3) instead of H2O.

P4L24: can you list the simulated intermediates here (if practical)?

Yes, we added two examples of the additional species.

**page 4:**

**Old:** The mechanism is extended to resolve specific intermediates in the CH4 → H2O reaction chain, resulting in slightly more comprehensive chemical kinetics.

**New:** The mechanism is extended to resolve specific intermediates in the CH4 → H2O reaction chain (e.g. methyl (CH3) and methoxy radical (CH3O)), resulting in slightly more comprehensive chemical kinetics.

P4L30: please define acronyms, e.g. NMHCs, HCFCs

Yes, of course. We added the definitions.

P4L30: how do you define HOx? H+OH+HO2?

We define HOx as the sum of OH and HO2, which is now also indicated in the text as well.

P5: Please also define NOx, ClOx and BrOx

> That is included now as well.

> P5L28: Id like to know more about your RC1SD-base-10 EMAC simulation. E.g., from the name, can it be inferred that dynamics are specified to a reanalysis?

> Yes, that is right. SD stands for specified dynamics. We added some more information on this simulation.

This simulation is carried out at T42L90MA resolution with specified dynamics, hence a Newtonian relaxation is performed with respect to meteorological reference data (ERA-Interim reanalysis data from ECMWF (Dee et al., 2011) to be more precise) concerning the prognostic variables divergence, vorticity, temperature and (logarithm of) surface pressure.

> Fig.2: where is H2O being lost to in the mesosphere?

> The reduction of H2O occurs in the model due to two chemical reactions and photolysis, as indicated by Reactions (R7-R9). We added a reference in the text to make this clear.

In the mesosphere the loss of $H_2O$ especially via Reactions (R7) and (R9) increases (also evident in Fig. 2).

> Table 2: the authors might want to consider adding an extra column stating whether the simulation is a box model or CCM simulation.

> That is a good idea. We added such a column.

> P15L21-23: Can you comment on what in particular is important regarding chemical composition of the box?

> The variations are mostly dependent on the abundances of the atmospheric radicals (OH, HO2) and on the (additional) reaction partners of $CH_4$ ($O(^1D)$ and Cl). We added such an explanation.

**page 15:**

**Old:** Consideration of the obvious vertical dependence and the very low temperature dependence gives evidence that not the physical parameters (temperature and pressure) itself are crucial for the $H_2O$ yield, but rather the chemical composition of the box. This chemical composition, however, changes with altitude (hence with pressure) and depends additionally on transport

**New:** Consideration of the obvious vertical dependence and the very low temperature dependence gives evidence that not the physical parameters (temperature and pressure) themselves are crucial for the $H_2O$ yield, but rather the chemical composition of the box (i.e., among others, abundances of OH, $HO_2$, $O(^1D)$ and Cl). This chemical composition, however, changes with altitude (hence with pressure) and depends additionally on transport.

> P24L13: I think its now fairly well recognised that online chemistry is necessary in many respects, e.g. the Southern Hemisphere circulation response to CO2 via O3 changes (Chiodo and Polvani, 2017).

> We thank the referee for this comment. We added a note and the citation to relate our argument to the current scientific knowledge.

**page 24:**

**Old:** This raises the question, whether a simplified parameterization of $\gamma_{H_2O}$ is indeed applicable for future climate projections or if it is necessary to simulate the full-chemistry, if an accurate SWV is desired.

**New:** This raises the question, whether a simplified parameterization of $\gamma_{H_2O}$ is indeed applicable for future climate projections, or if it is necessary to simulate the full-chemistry for an accurate representation of SWV. The need of on-line chemistry for meaningful climate projections has anyway already been shown e.g. by Chiodo and Polvani (2017) for a realistic response of SH circulation to $CO_2$ changes.

---

## Author Comment (AC3) · 4 May 2018

**Answer to the referee # 3**

**May 4, 2018**

Dear referee,
we thank you for the comments on our manuscript, which we answer point-by-point below. Page numbers relate to the discussion paper.

**Major comments**

> The most important thing: reading this paper might lead the reader to conclude that the assumption that dCH4/dH2O = 2 is not a good one. In fact, we have many observations (they are referenced in this paper) that show it is an excellent assumption throughout most of the stratosphere. I agree that the assumption breaks down at high altitudes.

Yes, indeed, we show that assuming a constant $\gamma_{H_2O} = \frac{\frac{d}{dt}[H_2O]}{\frac{d}{dt}[CH_4]} = 2$ is not suitable, because $\gamma_{H_2O}$ changes with altitude depending on the chemical regime and OH abundance.

As explained in the introduction (page 2, lines 19–30), observations can not distinguish the $H_2O$ produced by $CH_4$ oxidation from that by oxdiation of $H_2$, which is produced in the troposphere and transported into the stratosphere. This is explicitly stated by Hurst et al. (1999):

"The quantity $P_{H_2O}/L_{CH_4}$ is often erroneously referred to as the "yield" of $H_2O$ from $CH_4$ oxidation, even though it includes significant $H_2O$ production from $H_2$ oxidation. Since $> 95\%$ of $H_2$ present in the lower stratosphere originated in the troposphere, the oxidation of $H_2$ produced by stratospheric $CH_4$ oxidation (1)–(5a) represents $< 5\%$ of $L_{H_2}$. Hence it is not possible to directly determine a true yield of $H_2O$ from $CH_4$ oxidation using observations of $H_2O$ and $CH_4$, and the slope of the correlation between $H_2O$ and $CH_4$ ($\Delta H_2O/\Delta CH_4$) is simply $P_{H_2O}/L_{CH_4}$."

We examine the contribution by $CH_4$ oxidation and how it can potentially be represented in CCMs. In particular, if one wants to apply a paramterization like: $\frac{d}{dt}[H_2O] = \gamma_{H_2O} \cdot \frac{d}{dt}[CH_4]$, one must be aware not to mix $\gamma_{H_2O}$ with the yield from the oxidation of $H_2$, originating from the troposphere. We added a note to stress this on page 3.

**Old:** In the EMAC model (Jöckel et al., 2010), for example, explicitly configured in a CTM-like set-up without interactive chemistry, the production of SWV from $CH_4$ oxidation is calculated in a simplified way using a specifically introduced $CH_4$ tracer (by applying the CH4 submodel) according to:

$$\frac{d}{dt}[H_2O] = -\gamma_{H_2O} \cdot \frac{d}{dt}[CH_4] \tag{1}$$

with $\gamma_{H_2O} = 2$ as the yield of $H_2O$.

**New:** In the EMAC model (Jöckel et al., 2010), for example, explicitly configured in a CTM-like set-up without interactive chemistry, the production of SWV from $CH_4$ oxidation is calculated in a simplified way using a specifically introduced $CH_4$ tracer (by applying the CH4 submodel) according to:

$$\frac{d}{dt}[H_2O] = -\gamma_{H_2O} \cdot \frac{d}{dt}[CH_4] \tag{2}$$

with $\gamma_{H_2O} = 2$ as the yield of $H_2O$. Note, that if one wants to apply such a parameterization, one must specifically be aware not to mix $\gamma_{H_2O}$ with the yield from the oxidation of $H_2$, originating from the troposphere.

The reason the assumption is good in the lower stratosphere, even though the calculated yield there is less than 2, is that the lifetime of CH4 there is very long (100 years). Almost all of the oxidation of methane in the stratosphere is occurring in the mid-stratosphere, where the yield is 2. This air is transported down into the lower stratosphere, so the yield in the lower stratosphere just reflects mid-stratosphere photochemistry.
This needs to be clearly laid out in the paper. Otherwise, readers will be misinformed.

We agree on the lifetime. However, we do not see a downward transport of air in the tropical stratosphere for which we present our analysis. In contrast, the BDC transports air upward in that region (otherwise no tape recorder (Mote, 1995) would be visible).

Assessing the quality of the assumption that dH2O/dCH4 = 2 would require a different analysis. All one would have to do is show regressions of H2O versus CH4 in various regions of the stratosphere (from either observations or models with full stratospheric chemistry). This comparison would show you if that assumption is good.

This is the method usually applied to observations. However, as discussed above and in our manuscript, by using the correlations of $H_2O$ and $CH_4$ alone, it is not possible to distinguish between $H_2O$ from $CH_4$ oxidation, $H_2O$ from oxidized $H_2$, which is produced in the troposphere and transported to the stratosphere.

In fact, the paper is really about H photochemistry, not the assumption that dH2O/dCH4 = 2. There's a lot of discussion in the paper that revolves around the details of stratospheric photochemistry. So one possible suggestion that I think would improve the paper would be to remove the present motivation of the paper (testing if dH2O/dCH4 = 2) and replace it with a more accurate characterization of the work described (investigating H photochemistry and sensitivities).

Following our comment above, we are interested in the contribution of methane oxidation to SWV and how it is (or could be) represented in GCMs without a full photochemical mechanism. For this, $\gamma_{H_2O}$ is required, because we need to distinguish the above indicated contributions. In order to derive this $\gamma_{H_2O}$, we apply a CCM (i.e. with detailed photochemistry) as a reference because the simple tracer-tracer correlation method cannot provide the $\gamma_{H_2O}$ as we define it in our introduction (pages: 6–8).

**Smaller comments**

1) I would eliminate Fig. 1 below 100 hPa. This region is not relevant to the paper.

We are hesitating to cut the figures below 100 hPa for mainly two reasons: (1) the gradient accross the tropopause would not be visible anymore and (2) the consistently calculated tropoheric yield values, although not discussed, still provide valid information and serve as a reference for similar follow up studies.

2) Why do the authors spend so much time looking at OH sensitivity? That section should be motivated better.

Thank you for this suggestion. We invested some more sentences on the motivation of this section.

**Old:** The results of the previous section revealed that the effective yield of water vapor from $CH_4$ oxidation depends on the box location, hence the chemical regime at a certain pressure level. Particularly, OH is one of the major oxidants that largely controls the conversion of $CH_4$ to $H_2$ and $H_2O$ respectively.

In the simulations shown above (Exp1) the OH is unconstrained, however, its final (equilibrated) OH concentration does not deviate much from the initial values (see Fig. 5).

**New:** The results of the previous section reveal that the effective yield of water vapor from $CH_4$ oxidation depends on the box location, hence the chemical regime at a certain pressure level. Particularly, OH is one of the major oxidants shaping the chemical regime and largely controls the conversion of $CH_4$ to $H_2$ and $H_2O$, respectively. In the simulations shown above (Exp1) the OH is unconstrained, however, its final (equilibrated) OH concentration does not deviate much from the initial values (see Fig. 5). The following sensitivity study aims towards understanding the relationship between the OH and the vertical profile of the yield. It is investigated whether the variations of the yield are directly related to OH variations or to other parameters.

3) I don't understand why the direct and effective yields of water vapor in the lower stratosphere are equal. The direct yield is the water vapor produced directly from methane oxidation. However, there's also a contribution from oxidation of H2 (lifetimes of CH4 and H2 are similar in the lower strat.). That would be included in the effective yield. Thus, the effective yield should be larger than the direct yield, right? I'm confused.

The direct yield indicates the water produced by methane oxidation on a direct pathway. Once water is produced, it also gets reduced and subsequently recycled. Precisely, an H atom, which was part of $CH_4$, migrates to $H_2O$ and further to some H-carrying species and potentially back to $H_2O$. The ultimate amount of $H_2O$ per oxidized $CH_4$ is the effective yield, i.e., just that part, on an equilibrated level, which stays in $H_2O$. The effective yield is therefore always smaller than or equal to the direct yield.
This cycle is sketched in Fig. 1. The arrows of $L_{CH_4}$ and $P_{H_2O}^I$ are indicating the direct yield and all arrows together the effective yield. We add this information to the caption in the revised manuscript.
The contribution of $H_2$ is in our case only considered, if – and only if – the $H_2$ was previously produced by $CH_4$ oxidation. Recall that by applying MECCA-TAG we are able to distinguish between H introduced into the system by $CH_4$ from that introduced by other species.
Direct and effective yield are equal, as long as the loss of $H_2O$ is negligible or the recycling is lossless. To avoid confusion, we added this sentence to the text of the yield definitions in the manuscript.

4) This emphasizes that I don't particularly understand the way the authors have defined effective and direct yield. It seems to me that direct yield should be production of water directly from methane oxidation and effective yield should be the direct production plus the yield of water vapor from H2 oxidation and minus the loss of H2O from photochemistry. Is this how they view their definitions? If so, they should perhaps re-phrase that part of the manuscript.

We hope that the answer above gave some additional explanation to the definition of direct and effective yield. Additional to that, we would like to stress here that the effective yield is not the sum of the direct production of $H_2O$ from $CH_4$ oxidation and $H_2$ oxidation minus the loss of $H_2O$ from photochemistry. We track the H atoms, which all have in common that their source is (only) $CH_4$. These H atoms can

> temporarily be part of $H_2$, but we are not counting oxidation of $H_2$ which is produced in the troposphere and transported into the stratosphere. However, we are accounting for hydrogen, which has been part of $CH_4$ produced $H_2O$, and has been recycled. The method takes care that no double counting takes place. We added some corresponding notes to the introduction of the method.

**Old:** In this particular case, we count the $H_2O$ molecules created from $CH_4$ oxidation pathways and are able to distinguish the H from $CH_4$ from the H of other sources ($H_2$, NMHCs, HCFCs etc.). However, those that further break down to other HOx (OH+$HO_2$) compounds (and subsequently produce $H_2O$ again) are counted separately. Overall, such an approach is the "online" approximation of the technique used by Lehmann (2004) and helps to avoid double-counting issues in yield derivation. Ultimately, we are able to quantify the fraction of H atoms populating the species of the complete ($CH_4 \rightarrow H_2O/H_2 \leftrightarrow HO_x$)-cycle, including their fractions recycled via $H_2O$.

**New:** In this particular case, we count the $H_2O$ molecules created from $CH_4$ oxidation pathways and are able to distinguish the H from $CH_4$ from the H of other sources ($H_2$, NMHCs, HCFCs, etc.). More specifically, we track the H atoms, which all have in common that their source is only $CH_4$. These H atoms can temporarily be part of $H_2$, but we are not counting oxidation of $H_2$, which is produced in the troposphere and transported into the stratosphere. However, we are accounting for hydrogen, which has been part of $CH_4$ produced $H_2O$, and which becomes recycled after depletion of $H_2O$. Hence, that part of $CH_4$ produced $H_2O$, which breaks down to other HOx (OH+$HO_2$) compounds (and subsequently produces $H_2O$ again) is counted separately. Overall, such an approach is the "online" approximation of the technique used by Lehmann (2004) and helps to avoid double-counting issues in yield derivation. Ultimately, we are able to quantify the fraction of H atoms populating the species of the complete ($CH_4 \rightarrow H_2O/H_2 \leftrightarrow HO_x$)-cycle, including their fractions recycled via $H_2O$.

**References**

Hurst, D., Dutton, G., Romashkin, P., Wamsley, P., Moore, F., Elkins, J., Hintsa, E., Weinstock, E., Herman, R., Moyer, E., Scott, D., May, R., and Webster, C.: Closure of the total hydrogen budget of the northern extratropical lower stratosphere, J. Geophys. Res. Atmos., 104, 8191–8200, 1999.

Jöckel, P., Kerkweg, A., Pozzer, A., Sander, R., Tost, H., Riede, H., Baumgaertner, A., Gromov, S., and Kern, B.: Development cycle 2 of the Modular Earth Submodel System MESSy2, Geosci. Model Dev., 3, 717–752, doi: 10.5194/gmd-3-717-2010, manual, 2010.

Lehmann, R.: An Algorithm for the Determination of All Significant Pathways in Chemical Reaction Systems, J. Atmos. Chem., 47, 45–78, doi: 10.1023/B:JOCH.0000012284.28801.b1, 2004.

Mote, P.: The annual cycle of stratospheric water vapor in a general circulation model, J. Geophys. Res., 100, 7363–7379, doi: 10.1029/94JD03301, URL http://onlinelibrary.wiley.com/doi/10.1029/94JD03301/pdf, 1995.

---

## Author Response (AR2)

**Answer to the Co-editor**

June 21, 2018

Dear Jens-Uwe Grooß,

according to your additional concerns we revised the manuscript. We answer your comments point by point below. We further added a document highlighting the changes in the manuscript.

> Dear Franziska Frank at al.,
>
> Thank you for yours answers and for preparing the revised version.
>
> I must say that I am still puzzled by some of your findings. Especially, I think you should address the critics of Reviewer #3 somewhat more who stressed the point that a lot of stratospheric observations support gamma_H2O=2 and this ratio would break down above the stratopause. Therefore I still have some questions regarding your study:

> As we stress in our revision le Texier et al. (1988) already showed that the yield (chemical/kinetic production) of $H_2O$ is below two. We re-evaluate these findings by (1) distinguishing between the direct and effective yields and (2) by applying a comprehensive chemical mechanism.
> Moreover, we clearly show that the ratio of the hydrogen carrying species does not reflect the yield as pointed out by Hurst et al. (1999) (see also reply to referee #3).

**major issues:**

(page numbers in answers correspond to the new revised manuscript)

> To my understanding, there is no process above the "freeze-drying" altitude, that would be able to change total water (H2 + H2O + 2*CH4). Orders of magnitude are 4-7 ppmv H2O, 0-2 ppmv CH4, and 0.5 ppmv H2. This is consistent with your figure 12. The H2 mixing ratio throughout the troposphere and the stratosphere is about constant at about 0.5 ppmv (likely due to approximate equally strong production and loss processes).
>
> Therefore, if the effective yield would be significantly below 2 and H2 stays approximately constant at ∼0.5 ppmv, the hydrogen atoms must be found in a significant amount in a different (intermediate?) species. This seems not the case in the results displayed in your Figure 12. Throughout the stratosphere (below ∼0.2 hPa) the sum of H2O + 2*CH4 is constant. For me that means that effectively all CH4 molecules are converted into 2 H2O molecules (potentially involving equal production and loss of H2).
>
> In other words, if there were no H2 contribution from the troposphere, you would see an increase in H2 with altitude due to CH4-oxidation with gamma_H2O < 2. So one may need to consider gamma_H2O and gamma_H2, but also the oxidation of H2 forming H2O.

> We largely agree. However, the contribution of $H_2$ transported originally from the troposphere at a given point in the stratosphere cannot be distinguished from that $H_2$, which is produced there chemically in the

[Figure]

Figure 12: Annual zonal average of tagged H content by species (in $\mu$mol mol$^{-1}$) over the tropics (23° S–23° N).

stratosphere (and as such defining the chemical yield), neither from observations, nor from our Figure 12.

The chemical yield of $H_2O$ from methane (both, direct and effective) is defined solely by chemical production of $H_2O$ (i.e., methane oxidation), which competes with the chemical production of $H_2$ (and H) from $CH_4$.

Thus, indeed the contribution of $H_2$ produced from methane increases with altitude (corresponding to $\gamma_{H_2O}$ <2), whereas the $H_2$ originally injected from the troposphere decreases (by oxidation into $H_2O$). Therefore the net $H_2$ content is (almost) constant, at least in the lower stratosphere where measurements are available.

In order to clarify this even further, we added an additional panel to Figure 12 (see also in this reply), which shows in addition the tagged species, i.e. those species, which carry H atoms and were originally part of a $CH_4$ molecule. The text has been expanded accordingly with an explanation.

**New (page 18):** The chemical regime determines the proportion between H, $H_2$ and $H_2O$, but the total H content is preserved. Figure 12 (right) shows the tagged H content in the same manner. In this panel the difference between the total $H_2$ including the transported $H_2$ from the troposphere, which is observed in atmospheric measurements, and the $H_2$ solely produced by $CH_4$ becomes distinguishable. The contribution of $H_2$ produced from methane increases with altitude (corresponding to $\gamma_{H_2O}$<2), whereas the $H_2$ originally injected from the troposphere decreases (by oxidation into $H_2O$). Therefore the net $H_2$ content is (almost) constant, at least in the lower stratosphere where measurements are available.

The question is also, at what altitudes this is important. A deviation from the factor 2 is clear above 0.2 hPa, but in the lower stratosphere it is (at least) not important. This is so because of the comparison between the altitude profiles of the effective yield (Fig. 3) and the CH4 loss rate or the CH4 lifetime (Fig. 4). If (in the lower stratosphere) the chemical lifetime is above the typical transport times or in other words the loss rate is very low, then the derived yield is not very relevant.

Thank you for pointing this out. Indeed, the yield alone does not say much about its relevance at a given altitude. In addition the contribution of $CH_4$ oxidation to total water needs to be considered as well. As can be seen from our updated Figure 12, the ratio of tagged $H_2O$ (i.e., those from methane, right panel) to total $H_2O$ (left panel) between 100 and 10 hPa is between 1 and 1.75 by 4, i.e. the contribution of $H_2O$ from $CH_4$ is in the range of 25% to 44%.

Our calculated yield at these altitudes is 1.5 to 1.8. Thus, the assumption of $\gamma_{H_2O} = 2$ overestimates this contribution by 10% (1.8/2) to 25% (1.5/2), which is equivalent to an overestimation of total water of 2.5% (0.1 * 0.25) up to 11% (0.25 * 0.44).

Given the large uncertainties of $H_2O$ measurements in this altitude range and the high sensitivity for climate impact (Solomon et al., 2010), a 10% change in water vapor can have a measurable impact. This impact can only be estimated by sensitivity climate simulations. These are, however, beyond the scope of our present study.

We added a brief discussion about the relevance to the revised manuscript.

**New (page 24):** Based on our simulations, in the lower stratosphere between 100 and 10 hPa, the portion of $H_2O$ from $CH_4$ is in the range of 25% to 44% (calculated by Fig. 12 taking the ratio of tagged and total $H_2O$). Assuming $\gamma_{H_2O} = 2$ overestimates the contribution of $CH_4$ oxidation to the $H_2O$ production by 10% to 25%, which is equivalent to an overestimation of total water of 2.5% up to 11%. Given the large uncertainties of $H_2O$ measurements in this altitude range and the high sensitivity for climate impact (Solomon et al., 2010), a 10% change in water vapor can have a measurable impact. This impact can only be estimated by sensitivity climate simulations. These are, however, beyond the scope of our present study.

Could it be that your effective yield is below 2 because you do not include all follow-up reactions of the CH4 oxidation? Or enough "recycling cycles"?

We are certain that this is not the case. We use a very comprehensive chemical mechanism and the tagging method takes care that the whole recycling process is considered.

Figure 10 shows the effective yield in the 3-D model. The values >2 are said to be due to transport of intermediate species. It is not clear, what these intermediate species would be? Is it H2? No other intermediate species seems to have long lifetimes and/or a mixing ratio above the ppb level.

There are several intermediates between $CH_4$ oxidation and $H_2O$ production (except for the reaction with OH, where at least one $H_2O$ is produced right away). The intermediates are $CH_3$, $CH_3O_2$, $CH_3O$, $CH_3OOH$, HCHO and $CH_3OH$ to name only a few. Their atmospheric lifetimes in the stratosphere range from a couple of hours to months. $H_2$ is of course the most important intermediate. The residence times of the intermediates combined is sufficient to be transported upward. Note, however, that the kinetic yield is per molecule and therefore does not depend on the absolute mixing ratio of the educt species. We added a list of intermediates to the text and point to the relevance of $H_2$.

**New (page 17):** First, the yield of $H_2O$ from $CH_4$ oxidation increases in the upper stratosphere and lower mesosphere to a value above 2, because the global model, unlike the box model, includes transport. The tagged intermediates (e.g. tagged $H_2$ (mainly), $CH_3$, $CH_3OH$, HCHO etc.) which are produced at lower levels are transported upward and are finally converted to $H_2O$. This results in a production of more than two $H_2O$ molecules per oxidized $CH_4$ in one specific layer, because the additional production via transported intermediates is counted as well. In layers, where this increased production takes place, high OH concentration supports the conversion of the intermediates towards $H_2O$, since OH is the main driver of the chemistry (e.g. $H_2 + OH \rightarrow H_2O + H$).

> Also that means that this yield would be dependent on the specific transport formulation.

> We think that we agree that specific transport formulations can hardly be considered in a parameterization of $\gamma_{H_2O}$, however, some of our formulations might be misleading.
> We therefore reformulated the text.

**New (page 25):** Besides this, transport of intermediates is an important factor for the vertical profile of the $\gamma_{H_2O}$. It must be noted that atmospheric transport is not constant in time. The Brewer-Dobson circulation, for example, changes in future climate projections (Butchart et al., 2010). A simple parameterization of $\gamma_{H_2O}$ cannot take these changes in transport into account, since they depend on various factors. This raises indeed the question, whether a simplified parameterization of $\gamma_{H_2O}$ is at all applicable for future climate projections, or if it is necessary to simulate the full-chemistry for an accurate representation of SWV. The need of on-line chemistry for meaningful climate projections has anyway already been shown e.g. by Chiodo and Polvani (2017) for a realistic response of SH circulation to $CO_2$ changes.

> I appreciate that you included the subsection 4.1 for GCM recommendations. However I fear there is not much use for GCM modellers. The idea of a pressure-dependent gamma_H2O is likely not sufficient, as I think, it should at least depend also on latitude. It seems difficult to use a yield that is dependent on the specific transport formulation. Would be better to involve the results from the Lagrangian results that are displayed in figs 2 and 5 and a simplified system involving CH4–>H2, CH4–>H2O and H2–>H2O? This would do the transport of theses threee species consistently with the others.

> Although we show (in the supplement) that our yield results are hardly dependent on latitude (except for the polar regions), we added latitude as a potential parameter for a simple parameterization for non-climate change simulations. Additionally, we added a statement about the feasibility of a three tracer parameterization.
> We are however confused by your statement about the Lagrangian results since we do not apply a Lagrangian model. We apply a box model. The boxes are fixed in space and not transported.

**New (page 25):** Keeping these challenges in mind we are interested in deriving a parameterization as an intermediate stage between the very simple constant yield and the on-line chemistry. This is beyond the scope of the current study. Nevertheless, in the paragraph below we provide a sketch of such a parameterization together with its limitations and requirements.

One could start with a parameterization as introduced by Eq. (1), however, with a pressure $p$ (and latitude $\phi$) dependent $\gamma_{H_2O}(p, \phi)$ derived from our vertical yield profiles. This adds a vertical dependency to the chemical production of $H_2O$ per $CH_4$ oxidized. As long as no large variations or trends in the stratospheric transport are expected within the simulation period, our profile is a good approximation. The limitation is, however, that the pressure (and latitude) dependence is likely to change with changing climate.

> Reviewer #1 asked you to include the results from the study by Wrotny et al. (2010) who show a derived yield of 2.3 at 4.6 hPa while in your Figure 10 the yield is about 1.8 at this pressure level. Could you comment on that?

> Our results are lower at most pressure levels investigated by Wrotny et al. (2010). As they stated in their conclusions "the net loss of $H_2$ [...] drives additional $H_2O$ production, thus producing positive vertical gradients in $H_2O+2*CH_4$" (Wrotny et al., 2010). In other words, they attribute the values above 2 to the production from $H_2$. Our method distinguishes $H_2$ produced by $CH_4$ oxidation from $H_2$ from other sources (e.g. transport from the troposphere) and our yield is only defined for the $CH_4$ originating part.

> Therefore it is lower than reported by Wrotny et al. (2010).
> We added this explanation to our text.

**Old:** This is furthermore consistent with the findings of Wrotny et al. (2010), who calculated a yield larger than 2 in this area as well.

**New:** This is furthermore consistent with the findings of Wrotny et al. (2010), who calculated a yield larger than 2 in this area as well. However, our results are lower than from Wrotny et al. (2010). As they stated in their conclusions "the net loss of $H_2$ [...] drives additional $H_2O$ production, thus producing positive vertical gradients in $H_2O+2*CH_4$" (Wrotny et al., 2010). In other words, they attribute the values above 2 to the production from $H_2$. Our method distinguishes $H_2$ produced by $CH_4$ oxidation from $H_2$ from other sources (e.g. transport from the troposphere) and our yield is only defined for the $CH_4$ originating part. Therefore it is lower than reported by Wrotny et al. (2010).

**minor issues:**

(page and line numbers from the version with highlighted changes)

> p.18 l.13. I think you cannot assume that the proportion of H / total hydrogen is constant. H does have a clear diurnal cycle while total hydrogen does not

> Yes, that is true. There are also monthly variations in the sum of H, $H_2$ and $H_2O$. However, we average over sufficiently long time periods (i.e. a year). We are therefore not affected by these variations. For the very same reason the argumentation in the half-sentence actually becomes obsolete. We reformulated the sentence accordingly.

**Old:** If we assume further that the simulated proportion of H, $H_2$ and $H_2O$ at a certain level is approximately constant in time and that $CH_4$ is at higher layers the only additional hydrogen supply, we can determine the effective yield of $H_2O$ by $CH_4$ oxidation through the proportion of H atoms in $H_2O$ to the total hydrogen content of H, $H_2$ and $H_2O$.

**New:** If we assume further that $CH_4$ is at higher layers the only additional hydrogen supply, we can determine the effective yield of $H_2O$ by $CH_4$ oxidation through the proportion of anually averaged H atoms in $H_2O$ to the total hydrogen content of H, $H_2$ and $H_2O$.

> figure 13. It is unclear what are you showing. Is it something like d(H2O-tagged)/d CH4 and d(H + 2H2 + 2H2O)/d CH4 ? please clarify.

> Thank you for pointing this out. It is the proportion of H in tagged and total $H_2O$ to the sum of H, $H_2$ and $H_2O$. I.e.:
> $$\frac{2 \cdot H_2O}{H + 2 \cdot H_2 + 2 \cdot H_2O} \text{ and } \frac{2 \cdot H_2O_{tagged}}{H_{tagged} + 2 \cdot H_{2tagged} + 2 \cdot H_2O_{tagged}}.$$
> We added this explanation to the figure caption.

> discussion 3rd method with OH constrained: If you use a constant OH value that may be inconsistent with the remaining chemical composition, total hydrogen would be not conserved. Is this a problem or do you see deviations in total hydrogen itself in addition to the H2O yield?

We checked whether the constant OH introduces a concerning amount of hydrogen to the system. This is not the case. The total hydrogen in the system is not influenced by keeping OH constant. This is already mentioned on page 9: "In further sensitivity simulations with CAABA, OH is initialized with the reference from EMAC multiplied with constants and kept constant throughout the simulation. This introduces an additional prescribed hydrogen carrying species, which introduces or withdraws hydrogen to or from the system. However, contribution of OH to the total H abundance in the system was found negligible."

**typographical issues:**

(page and line numbers from the version with highlighted changes)

p. 19 line 3: you likely mean "total hydrogen content"

Thank you! We changed it.

fig 12, legend blue dashed should be "H + 3 umol mol$^{-1}$" (alternatively you could use a x-range from -1 to 18 and display H without shift)

That is a good idea. We changed the x-range starting with -1.

**References**

[revised manuscript text omitted]

<

---

## Author Response (AR3)

**Answer to the Co-editor**

**July 2, 2018**

Dear Jens-Uwe Grooß,

thank you for your comments. We appreciate your concerns and feel that this truly improves the manuscript. You find our answers to your comments below.

**minor revisions**

> Looking on your figure 10 at 0.3 hPa there is yield of 2.2, that means 10% more H2O production compared with the CH4 loss rate. At this altitude, I read from your figure 12 about 0.25 ppmv CH4, 0.5 ppmv H2 and 6.5 ppmv H2O, figure 4 gives a CH4 lifetime of $\sim$0.8y.
> Therefore, the CH4 loss is in the order of magnitude of 0.3 ppmv/y and the imbalance must be about 0.03 ppmv/y. The shorter the lifetime of an intermediate species, the larger mus be the concentration in order to create this amount of imbalance by transport, i.e. 0.06 ppmv, 0.36 ppmv and 1 ppmv for a lifetime of 6 months, 1 month and 10 days, respectively. Please clarify, what the intermediate species this would be.

> Thank you for this comment. We are indeed sure that tagged $H_2$, with its comparable lifetime and abundance to that of $CH_4$ ($\sim$0.25 ppmv), is the intermediate, which is of most concern here. We sharpened the argument on page 17 (line 17–23) and added specifications concerning the intermediate $H_2$ molecule at other points in the manuscript as well.

**Old (page 17):** First, the yield of $H_2O$ from $CH_4$ oxidation increases in the upper stratosphere and lower mesosphere to a value above 2, because the global model, unlike the box model, includes transport. The tagged intermediates (e.g. tagged $H_2$ (mainly), $CH_3$, $CH_3OH$, HCHO etc.) which are produced at lower levels are transported upward and are finally converted to $H_2O$. This results in a production of more than two $H_2O$ molecules per oxidized $CH_4$ in one specific layer, because the additional production via transported intermediates is counted as well. In layers, where this increased production takes place, high OH concentration supports the conversion of the intermediates towards $H_2O$, since OH is the main driver of the chemistry (e.g. $H_2 + OH \rightarrow H_2O + H$).

**New (page 17):** First, the yield of $H_2O$ from $CH_4$ oxidation increases in the upper stratosphere and lower mesosphere to a value above 2, because the global model, unlike the box model, includes transport. The tagged intermediates which are produced at lower levels are transported upward and are finally converted to $H_2O$. The by far most abundant intermediate is tagged $H_2$ with about 0.25 $\mu$mol mol$^{-1}$ at 0.2 hPa. Hydrogen gas has a comparably as long lifetime as $CH_4$, since it is reacting with the same species and comparable reaction rates (Rahn et al., 2003; Sander et al., 2011). The transported tagged $H_2$ reacts further to $H_2O$. This results in a production of more than two $H_2O$ molecules per oxidized $CH_4$ in one specific layer, because this additional production is counted as well. In layers, where the increased production takes place, high OH concentration supports the conversion of the intermediates towards $H_2O$, since OH is the main driver of the chemistry (e.g. $H_2 + OH \rightarrow H_2O + H$).

new figure 12b: Why does tagged H2O have already about 2.6 ppmv H2O in the troposphere? Does that mean, that already 1.3 ppmv of CH4 is already oxidized to H2O?

In the troposphere this type of calculation remains challenging, since $H_2O$ is part of the hydrological cycle. We would claim that even more than 2.6 ppmv is produced, but is to a large extent removed through condensation and sedimentation. Note, that over 90% of all $CH_4$ emitted at the surface is removed in the troposphere. What we indeed derive from Figure 12b is that in the troposphere all hydrogen from $CH_4$ is oxidized to $H_2O$. The reason for this is the fast kinetic reaction chain under tropospheric conditions. Still, the chemically produced $H_2O$ in the troposphere is negligible compared to the overall tropospheric humidity, which is up to 3 magnitudes larger.

**References**

[revised manuscript text omitted]

<